# Mildly Conservative $Q$-Learning for Offline Reinforcement Learning

**Jiafei Lyu**[1][*] **Xiaoteng Ma**[2][*], **Xiu Li**[1][†] **Zongqing Lu**[3][†]

[1]Tsinghua Shenzhen International Graduate School, Tsinghua University
[2]Department of Automation, Tsinghua Unversity
[3]School of Computer Science, Peking University
{lvjf20,ma-xt17}@mails.tsinghua.edu.cn,
li.xiu@sz.tsinghua.edu.cn, zongqing.lu@pku.edu.cn

## Abstract

Offline reinforcement learning (RL) defines the task of learning from a static logged dataset without continually interacting with the environment. The distribution shift between the learned policy and the behavior policy makes it necessary for the value function to stay conservative such that out-of-distribution (OOD) actions will not be severely overestimated. However, existing approaches, penalizing the unseen actions or regularizing with the behavior policy, are too pessimistic, which suppresses the generalization of the value function and hinders the performance improvement. This paper explores mild but enough conservatism for offline learning while not harming generalization. We propose Mildly Conservative $Q$-learning (MCQ), where OOD actions are actively trained by assigning them proper pseudo $Q$ values. We theoretically show that MCQ induces a policy that behaves at least as well as the behavior policy and no erroneous overestimation will occur for OOD actions. Experimental results on the D4RL benchmarks demonstrate that MCQ achieves remarkable performance compared with prior work. Furthermore, MCQ shows superior generalization ability when transferring from offline to online, and significantly outperforms baselines. Our code is publicly available at https://github.com/dmksjfl/MCQ.

## 1 Introduction

Continually interacting with the environment of online reinforcement learning (RL) is often infeasible and unrealistic, since the data collection process of the agent may be expensive, difficult, or even dangerous, especially in real-world applications. Offline RL, instead, aims at learning from a static dataset that was previously collected by some unknown process [36], hence eliminating the need for environmental interactions during training.

The main challenge of offline RL is the distribution shift of state-action visitation frequency between the learned policy and the behavior policy. The evaluation of out-of-distribution (OOD) actions causes extrapolation error [14], which can be exacerbated through bootstrapping [34] and result in severe overestimation errors. Thus, keeping conservatism in value estimation is necessary in offline RL [24, 50, 60]. Previous methods achieve the conservatism by compelling the learned policy to be close to the behavior policy [14, 58, 34, 13, 57], by penalizing the learned value functions from being over-optimistic upon out-of-distribution (OOD) actions [35, 33, 59], or by learning without querying OOD samples [56, 8, 62, 32, 40].

---

[*]Equal Contribution
[†]Corresponding Authors

36th Conference on Neural Information Processing Systems (NeurIPS 2022).

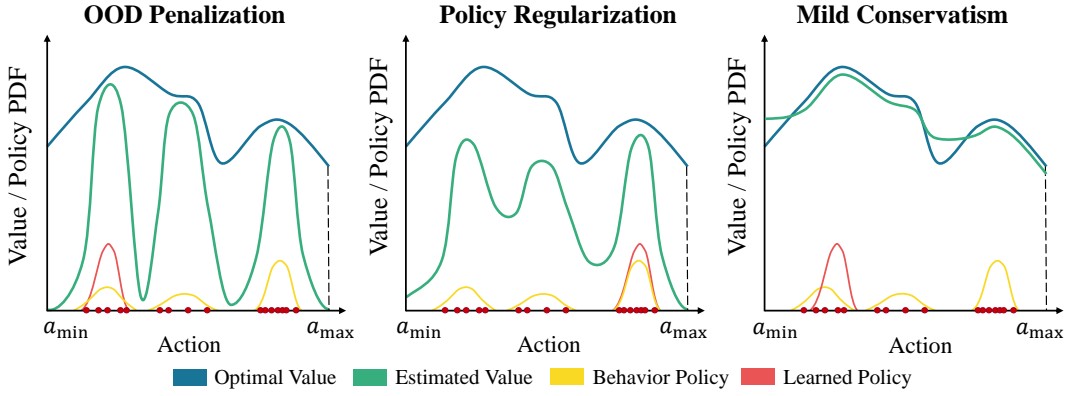

Figure 1: Comparison of prior methods against mild conservatism. The red spots represent the dataset samples. The left figure shows that penalizing OOD actions makes the value function drop sharply at the boundary of the dataset's support, which barriers policy learning. The central figure depicts that policy regularization keeps the policy near behavior policy, leading to undesired performance if the behavior policy is unsatisfying. On the right side, we illustrate the basic idea of mild conservatism. The estimated values for OOD actions are allowed to be high as long as it does not affect the learning for the optimal policy supported by the dataset, i.e., $Q(s, a^{\mathrm{ood}}) < \max_{a \in \mathrm{Support}(\mu)} Q(s, a)$.

In practice, we rely on neural networks to extract knowledge from the dataset and generalize it to the nearby unseen states and actions when facing continuous state and action spaces. In other words, we need the networks to "stitch" the suboptimal trajectories to generate the best possible trajectory supported by the dataset. Unfortunately, there is no free lunch. Conservatism, which offline RL celebrates, often limits the generalization and impedes the performance of the agent. Existing approaches are still inadequate in balancing conservatism and generalization. As illustrated in Figure 1, policy regularization is unreliable for offline RL when the data-collecting policy is poor, and value penalization methods often induce unnecessary pessimism in both the in-dataset region and OOD region. We argue that *the proper conservatism should be as mild as possible*. As depicted in Figure 1, we aim at well estimating the value function in the support of the dataset, and allowing value estimates upon OOD actions to be high (even higher than their optimal values) as long as $Q(s, a^{\mathrm{ood}}) < \max_{a \in \mathrm{Support}(\mu)} Q(s, a)$ is satisfied. The mild conservatism benefits generalization since value estimates upon OOD actions are slightly optimistic instead of being overly conservative.

To fulfill that, we propose a novel _Mildly Conservative Bellman_ (MCB) operator for offline RL, where we *actively* train OOD actions and query their $Q$ values. We theoretically analyze the convergence property of the MCB operator under the tabular MDP setting. We show that the policy induced by the MCB operator is guaranteed to behave better than the behavior policy, and can consistently improve the policy with a tighter lower bound compared with policy constraint methods or value penalization methods like CQL [35]. For practical usage, we propose the practical MCB operator and illustrate its advantages by theoretically showing that *erroneous overestimation error will not occur* with it. We then estimate the behavior policy with a conditional variational autoencoder (CVAE) [29, 51], and integrate the practical MCB operator with the Soft Actor-Critic (SAC) [20] algorithm. To this end, we propose our novel offline RL algorithm, Mildly Conservative $Q$-learning (MCQ).

Experimental results on the D4RL MuJoCo locomotion tasks demonstrate that MCQ surpasses recent strong baseline methods on most of the tasks, especially on non-expert datasets. Meanwhile, MCQ shows superior generalization capability when transferring from offline to online, validating our claims that mild pessimism is of importance to offline learning.

## 2 Preliminaries

We consider a Markov Decision Process (MDP) specified by a tuple $\langle \mathcal{S}, \mathcal{A}, r, \rho_0, p, \gamma \rangle$, where $\mathcal{S}$ is the state space, $\mathcal{A}$ is the action space, $r(s, a) : \mathcal{S} \times \mathcal{A} \mapsto \mathbb{R}$ is the reward function, $\rho_0(s)$ is the initial state distribution, $p(s'|s, a) : \mathcal{S} \times \mathcal{A} \times \mathcal{S} \mapsto [0, 1]$ is the transition probability, $\gamma \in [0, 1)$ is the discount

factor. Reinforcement learning (RL) aims at finding a policy $\pi(\cdot|s)$ such that the expected cumulative long-term rewards $J(\pi) = \mathbb{E}_{s_0 \sim \rho_0(\cdot), a_t \sim \pi(\cdot|s_t), s_{t+1} \sim p(\cdot|s_t, a_t)}[\sum_{t=0}^{\infty} \gamma^t r(s_t, a_t)]$ are maximized. The state-action function $Q(s, a)$ measures the discounted return starting from state $s$ and action $a$, and following the policy $\pi$. We assume that the reward function $r(s, a)$ is bounded, i.e., $|r(s, a)| \leq r_{\max}$. Given a policy $\pi(\cdot|s)$, the Bellman backup for obtaining the corresponding $Q$ function gives:

$$\mathcal{T}^{\pi}Q(s, a) := r(s, a) + \gamma \mathbb{E}_{s'} \mathbb{E}_{a' \sim \pi(\cdot|s')}[Q(s', a')]. \tag{1}$$

The $Q$ function of the optimal policy satisfies the following Bellman optimal operator:

$$\mathcal{T}Q(s, a) := r(s, a) + \gamma \mathbb{E}_{s'} \left[ \max_{a' \in \mathcal{A}} Q(s', a') \right]. \tag{2}$$

In offline RL setting, the online interaction is infeasible, and we can only have access to previously collected datasets $\mathcal{D} = \{(s_i, a_i, r_i, s'_{i+1}, d_i)\}_{i=1}^{N}$, where $d$ is the done flag. We denote the behavior policy as $\mu(\cdot|s)$. The Bellman backup relies on actions sampled from the learned policy, $a' \sim \pi(\cdot|s')$. However, $a'$ can lie outside of the support of $\mu$ due to the distribution shift between $\pi$ and $\mu$. The value estimates upon $a'$ can then be arbitrarily wrong, resulting in bad policy training. Unlike prior work, we *actively* train OOD actions by constructing them pseudo target values. In this way, we retain pessimism while enjoying better generalization.

# 3 Mildly Conservative $Q$-Learning

In this section, we first formally define the MCB operator and characterize its dynamic programming properties in the tabular MDP setting. We further give a practical version of the MCB operator. We show that no erroneous overestimation will occur with the MCB operator. Finally, we incorporate the MCB operator with SAC [20] and present our novel offline RL algorithm.

## 3.1 Mildly Conservative Bellman (MCB) Operator

**Definition 1.** *The Mildly Conservative Bellman (MCB) operator is defined as*

$$\mathcal{T}_{\text{MCB}}Q(s, a) = (\mathcal{T}_1 \mathcal{T}_2)Q(s, a), \tag{3}$$

*where*

$$\mathcal{T}_1 Q(s, a) = \begin{cases} Q(s, a), & \mu(a|s) > 0. \\ \max_{a' \sim \text{Support}(\mu(\cdot|s))} Q(s, a') - \delta, & else. \end{cases} \tag{4}$$

$$\mathcal{T}_2 Q(s, a) = \begin{cases} r(s, a) + \gamma \mathbb{E}_{s'} \left[ \max_{a' \in \mathcal{A}} Q(s', a') \right], & \mu(a|s) > 0, \\ Q(s, a), & else. \end{cases} \tag{5}$$

The basic idea behind this novel operator is that if the learned policy outputs actions that lie in the support region of $\mu$, then we go for backup; while if OOD actions are generated, we deliberately replace their value estimates with $\max_{a' \sim \text{Support}(\mu(\cdot|s))} Q(s, a') - \delta$, where $\delta > 0$ can be arbitrarily small. That is, different from standard Bellman backup, we set up a *checking procedure* (i.e., $\mathcal{T}_1$) of whether the previous backup (i.e., $\mathcal{T}_2$) involves OOD actions for the update. Intrinsically, we construct pseudo target values for OOD actions. We subtract a small positive $\delta$ such that OOD actions will not be chosen when executing policy via $\arg \max_{a \in \mathcal{A}} Q(s, a)$.

For a better understanding of the MCB operator, we theoretically analyze its dynamic programming properties in the tabular MDP setting. All proofs are deferred to Appendix A.

**Proposition 1.** *In the support region of the behavior policy, i.e.,* $\text{Support}(\mu)$*, the MCB operator is a $\gamma$-contraction operator in the $\mathcal{L}_{\infty}$ norm, and any initial $Q$ function can converge to a unique fixed point by repeatedly applying $\mathcal{T}_{\text{MCB}}$.*

**Proposition 2** (Behave at least as well as behavior policy)**.** *Denote $Q_{\text{MCB}}$ as the unique fixed point acquired by the MCB operator, then in $\text{Support}(\mu)$ we have: $Q_{\mu} \leq Q_{\text{MCB}} \leq Q_{\mu^*}$, where $Q_{\mu}$ is the $Q$ function of the behavior policy and $Q_{\mu^*}$ is the $Q$ function of the optimal policy in the batch.*

Proposition 2 indicates that the policy induced by the MCB operator can behave at least as well as the behavior policy, and can approximate the optimal batch-constraint policy. Apart from this advantage, we further show that the MCB operator results in milder conservatism. We start by observing that

value penalization method, like CQL [35], guarantees that the learned value function $\hat{Q}^\pi(s, a)$ is a lower bound of its true value $Q^\pi(s, a)$. It is also ensured that following such conservative update leads to a safe policy improvement, i.e., $J(\pi_{\mathrm{CQL}}) \geq J(\mu) - \mathcal{O}(\frac{1}{(1-\gamma)^2})$ (Theorem 3.6 in [35]). For explicit policy constraint methods, e.g., TD3+BC [13], the learned policy $\pi_p$ mimics the behavior policy $\mu$, and can hardly behave significantly better than $\mu$. We show in Proposition 3 that explicit policy constraint methods also exhibit a safe policy improvement, $J(\pi_p) \geq J(\mu) - \mathcal{O}(\frac{1}{(1-\gamma)^2})$, while the MCB operator can consistently improve the policy with a tighter lower bound.

**Proposition 3** (Milder Pessimism)**.** *Suppose there exists an explicit policy constraint offline rein-forcement learning algorithm such that the KL-divergence of the learned policy $\pi_p(\cdot|s)$ and the behavior policy $\mu(\cdot|s)$ is optimized to guarantee* $\max\left(\mathrm{KL}(\mu, \pi_p), \mathrm{KL}(\pi_p, \mu)\right) \leq \epsilon, \forall s$. *Denote* $\epsilon_\mu^{\pi_p} = \max_s |\mathbb{E}_{a \sim \pi_p} A^\mu(s, a)|$, *where $A^\mu(s, a)$ is the advantage function. Then*

$$J(\pi_p) \geq J(\mu) - \frac{\sqrt{2}\gamma\epsilon_\mu^{\pi_p}}{(1-\gamma)^2}\sqrt{\epsilon}, \tag{6}$$

*while for the policy $\pi_{\mathrm{MCB}}$ learned by applying the MCB operator, we have*

$$J(\pi_{\mathrm{MCB}}) \geq J(\mu). \tag{7}$$

**In summary,** the MCB operator benefits the offline learning in two aspects: (1) the operator is a contraction, and any initial $Q$ functions are guaranteed to converge to a unique fixed point; (2) the learned policy of the MCB operator is ensured to be better than the behavior policy, and reserve milder pessimism compared with policy constraint methods or CQL.

### 3.2   Practical MCB Operator

In practice, it is intractable to acquire $\max_{a' \sim \mathrm{Support}(\mu(\cdot|s))} Q(s, a')$ in $\mathcal{T}_1$ of Eq. (4) in continuous control domains, and the behavior policy is often unknown. Thus, we fit an empirical behavior policy $\hat{\mu}$ with supervised learning based on the static dataset. The pseudo target values for the OOD actions are then computed by sampling $N$ actions from $\hat{\mu}$, and taking maximum over their value evaluation. Formally, we define the practical MCB operator below, accompanied by the theoretical analysis.

**Definition 2.** *The practical Mildly Conservative Bellman (MCB) operator is defined as*

$$\hat{\mathcal{T}}_{\mathrm{MCB}}Q(s, a) = (\hat{\mathcal{T}}_1\mathcal{T}_2)Q(s, a), \tag{8}$$

*where*

$$\hat{\mathcal{T}}_1 Q(s, a) = \begin{cases} Q(s, a), & \mu(a|s) > 0. \\ \mathbb{E}_{\{a_i'\}^N \sim \hat{\mu}(\cdot|s)}\left[\max_{a' \sim \{a_i'\}^N} Q(s, a')\right], & else. \end{cases} \tag{9}$$

Compared with Eq. (4), we make a small modification of $\mathcal{T}_1$, and keep $\mathcal{T}_2$ unchanged. There is no need to subtract $\delta$ here as generally $\mathbb{E}_{\{a_i'\}^N \sim \hat{\mu}(\cdot|s)}\left[\max_{a' \sim \{a_i'\}^N} Q(s, a')\right] \leq \max_{a' \sim \mathrm{Support}(\mu)} Q(s, a')$. The practical MCB operator is much easier to implement in practice. We show that the practical MCB operator is still a $\gamma$-contraction in the support region of the behavior policy $\mu$.

**Proposition 4.** *Proposition 1 still holds for the practical MCB operator.*

Since we fit the empirical distribution $\hat{\mu}$ of the behavior policy $\mu$, there may exist a shift between $\hat{\mu}$ and $\mu$, especially when we represent the policy via neural networks. That suggests that OOD actions $a'$ can still be sampled from $\hat{\mu}$ such that $a' \notin \mathrm{Support}(\mu(\cdot|s))$. Our last main result reveals that *erroneous overestimation issue will not occur* with the aid of the practical MCB operator.

**Proposition 5** (No erroneous overestimation will occur)**.** *Assuming that* $\sup_s D_{\mathrm{TV}}(\hat{\mu}(\cdot|s) \parallel \mu(\cdot|s)) \leq \epsilon < \frac{1}{2}$, *we have*

$$\mathbb{E}_{\{a_i'\}^N \sim \hat{\mu}(\cdot|s)}\left[\max_{a' \in \{a_i'\}^N} Q(s, a')\right] \leq \max_{a' \in \mathrm{Support}(\mu(\cdot|s))} Q(s, a') + (1 - (1 - 2\epsilon)^N)\frac{r_{\max}}{1 - \gamma}.$$

**Remark:** This proposition generally requires a comparatively well-fitted empirical behavior policy $\hat{\mu}$. In practice, we model $\hat{\mu}$ with a CVAE. In most cases, CVAE can already fit the dataset well and guarantee a good performance. Whereas there may exist some situations, e.g., the dataset is highly

multi-modal, then one can replace the CVAE as the conditional GAN (CGAN) to better capture the different modes in the dataset as depicted in [61]. We believe generative models like CGAN will be a good choice by then.

Intuitively, the above conclusion says that if the empirical behavior policy $\hat{\mu}$ well fits $\mu$, i.e., $\epsilon$ is small enough, then regardless of how $\{a'_i\}^N$ are sampled, the pseudo target value will approximate the maximum $Q$-value within the dataset's support with high probability. The extrapolation error is under the scale of $(1 - (1 - 2\epsilon)^N)\frac{r_{\max}}{1-\gamma}$. We expect a good empirical behavior policy such that most of the actions sampled from it will be in-distribution. However, if $\epsilon$ is large, $N$ can act as a trade-off parameter. The smaller $N$ we use, the more conservative we are. Fortunately, we find empirically that our method performs well in a large interval of $N$ over different tasks (see Section 4.2). Hence, it is safe to fix a $N$ in practice.

## 3.3 Algorithm

As aforementioned, we often cannot get prior information about the behavior policy $\mu$. Thus, we need to empirically fit a behavior policy $\hat{\mu}$ with supervised learning for applying the practical MCB operator. Our algorithm, Mildly Conservative Q-learning (MCQ), trains an additional generative model, which is also adopted by many prior work [14, 17, 33, 67]. We build our novel offline algorithm upon an off-the-shelf off-policy online RL algorithm, Soft Actor-Critic (SAC) [20].

**Modelling the behavior policy with the CVAE.** We utilize a conditional variational autoencoder (CVAE) [29, 51, 14] to model the behavior policy $\mu$. Given a fixed logged dataset, the goal of the CVAE is to reconstruct actions conditioned on the states such that the reconstructed actions come from the same distribution as the actions in the dataset, i.e., $\mu(\cdot|s)$. That generally satisfies the assumption we make in Proposition 5. As concerned by [32], training a generative model like CVAE still may produce out-of-dataset actions, which leads to extrapolation error since undefined $Q$ values can be possibly queried. Prior methods, like BCQ [14], do not well address such issue. While for our algorithm, such concern is mitigated because overestimation error is actually under control as is guaranteed by Proposition 5.

The CVAE $G_\omega(s)$ parameterized by $\omega$ is made up of an encoder $E_\xi(s,a)$ and a decoder $D_\psi(s,z)$ parameterized by $\xi, \psi$ respectively, $\omega = \{\xi, \psi\}$. The CVAE is optimized by maximizing its variational lower bound, which is equivalent to minimizing the following objective function.

$$\mathcal{L}_{\text{CVAE}} = \mathbb{E}_{(s,a)\sim\mathcal{D}, z\sim E_\xi(s,a)} \left[ (a - D_\psi(s,z))^2 + \text{KL}\left(E_\xi(s,a), \mathcal{N}(0,\mathbf{I})\right) \right], \qquad (10)$$

where $\text{KL}(p, q)$ denotes the KL-divergence between probability distribution $p(\cdot)$ and $q(\cdot)$, and $\mathbf{I}$ is the identity matrix. When sampling actions from the CVAE, we first sample a latent variable $z$ from the prior distribution, which is set to be multivariate normal distribution $\mathcal{N}(0, \mathbf{I})$, and then pass it in conjunction with the state $s$ into the decoder $D_\psi(s,z)$ to get the desired decoded action.

It is also worth noting that we do not choose GAN [19] as the generative model because it is known to suffer from training instability and mode collapse [52, 6, 5]. Also, GAN consumes much more time and memories to train compared with the CVAE.

**Loss functions.** In deep RL, the $Q$ function is represented with a neural network parameterized by $\theta$ and is updated via minimizing the temporal difference (TD) loss $\mathbb{E}_{s,a,r,s'}[(Q_\theta(s,a) - \mathcal{T}Q(s,a))^2]$. We actually are performing the regression task $(s,a) \mapsto \mathcal{T}Q(s,a)$ to train the $Q$ function. The target value $\mathcal{T}Q(s,a)$ is usually computed by utilizing a lagging target network parameterized by $\theta'$ without gradient backpropagation. As a typical actor-critic [30, 31, 53] algorithm, SAC uses its critic networks to perform value estimation and uses a separate actor network for policy improvement. In order to incorporate the MCB operator with the off-the-shelf SAC algorithm, we need to check whether the sampled action $a' \sim \pi(\cdot|s)$ lies outside of the behavior policy's support, i.e., whether $\mu(a'|s) > 0$. However, such a criterion is not reliable, because the true behavior policy $\mu$ is unknown and it is difficult to examine whether $\mu(a'|s) > 0$ in practice. It is also problematic if we rely on the empirical behavior policy $\hat{\mu}$ to check whether $a'$ is OOD as $\hat{\mu}$ itself can produce OOD actions.

We then resort to constructing an auxiliary loss for OOD actions and integrating it with the standard Bellman error. Specifically, we sample $a^{\text{ood}}$ from the learned policy $\pi(\cdot|s^{\text{in}})$ based on the sampled state $s^{\text{in}} \sim \mathcal{D}$ from the dataset and assign them pseudo target values based on the practical MCB operator. Note that the superscript ood is used to distinguish from the in-dataset real actions, and $a^{\text{ood}}$ is not necessarily an OOD action. We remark that if $a^{\text{ood}} \in \text{Support}(\mu(\cdot|s))$, the pseudo $Q$

---

**Algorithm 1** Mildly Conservative $Q$-learning (MCQ)

---

1: Initialize CVAE $G_\omega$, critic networks $Q_{\theta_1}, Q_{\theta_2}$ and actor network $\pi_\phi$ with random parameters
2: Initialize target networks $\theta'_1 \leftarrow \theta_1, \theta'_2 \leftarrow \theta_2$ and offline replay buffer $\mathcal{D}$.
3: **for** $t = 1$ to $T$ **do**
4:     Sample a mini-batch $B = \{(s, a, r, s', d)\}$ from $\mathcal{D}$, where $d$ is the done flag
5:     Train CVAE via minimizing Eq. (10)
6:     Get target value: $y = r(s, a) + \gamma \left[ \min_{i=1,2} Q_{\theta'_i}(s', a') - \alpha \log \pi_\phi(a'|s') \right], a' \sim \pi_\phi(\cdot|s')$
7:     Sample $N$ actions from $\pi$ based on each $s$ and $s'$, set $s^{\text{in}} = \{s, s'\}$
8:     Compute the target value for the OOD actions via Eq. (13)
9:     Update critic $\theta_i$ with gradient descent via minimizing Eq. (11)
10:     Update actor $\phi$ with gradient ascent via Eq. (14)
11:     Update target networks: $\theta'_i \rightarrow \tau\theta_i + (1 - \tau)\theta'_i, i = 1, 2$
12: **end for**

---

value will not negatively affect the evaluation upon it, because in-distribution actions are still trained to approximate the optimal batch-constraint $Q$ value. In this way, we *actively* train both possible OOD actions and in-distribution actions simultaneously via *OOD sampling*. The resulting objective function for the critic networks is presented in Eq. (11).

$$\mathcal{L}_{\text{critic}} = \lambda \mathbb{E}_{(s,a,r,s')\sim\mathcal{D}} \left[ (Q_{\theta_i}(s, a) - y)^2 \right] + (1-\lambda)\mathbb{E}_{s^{\text{in}}\sim\mathcal{D}, a^{\text{ood}}\sim\pi} \left[ (Q_{\theta_i}(s^{\text{in}}, a^{\text{ood}}) - y')^2 \right], \quad (11)$$

where the target value for the in-distribution actions gives

$$y = r(s, a) + \gamma \left[ \min_{i=1,2} Q_{\theta'_i}(s', a') - \alpha \log \pi_\phi(a'|s') \right], \alpha \in \mathbb{R}_+, \quad (12)$$

which follows the standard target value of vanilla SAC. The hyperparameter $\lambda$ balances the in-distribution data training and OOD action training. Following the formulas of the practical MCB operator in Eq. (9), the pseudo target value for the OOD action is computed by:

$$y' = \min_{j=1,2} \mathbb{E}_{\{a'_i\}^N \sim \hat{\mu}} \left[ \max_{a' \sim \{a'_i\}^N} Q_{\theta_j}(s^{\text{in}}, a') \right]. \quad (13)$$

Note that we experimentally find that replacing the min operator with a mean operator does not raise much difference in performance. We hence take advantage of the min operator to fulfill the pseudo clipped double $Q$-learning for OOD actions.

The policy is then optimized by solving the following optimization problem:

$$\pi_\phi := \max_\phi \mathbb{E}_{s\sim\mathcal{D}, a\sim\pi_\phi(\cdot|s)} \left[ \min_{i=1,2} Q_{\theta_i}(s, a) - \alpha \log \pi_\phi(\cdot|s) \right]. \quad (14)$$

We detail the learning procedure of our MCQ in Algorithm 1. Different from [13], our method does not require normalization over states or value functions. The only change we make to the vanilla SAC algorithm is an extra auxiliary loss term (blue term in Eq. (11)) such that OOD actions are actively and properly trained. The additional critic loss term can also be plugged into other off-policy online RL algorithms directly. As an evidence, we combine the MCB operator with TD3 [15], yielding a deterministic version of MCQ. Please refer to Appendix B for more details.

## 4 Experiments

In this section, we first empirically demonstrate the effectiveness and advantages of our proposed MCQ algorithm on D4RL benchmarks [12]. We then conduct a detailed parameter study to show the hyperparameter sensitivity of MCQ. We also experimentally illustrate that the value estimation of MCQ will not incur severe overestimation and pessimistic value estimates are witnessed in practice. Finally, we show the superior offline-to-online fine-tuning benefits of MCQ on some MuJoCo datasets.

### 4.1 Results on MuJoCo Datasets

We experimentally compare our MCQ against behavior cloning (BC), SAC, and several recent strong baseline methods, CQL [35], UWAC [59], TD3+BC [13], and IQL [32], on D4RL [12] benchmarks.

We choose these methods as they typically represent different categories of model-free offline RL, i.e., CQL is a value penalization method, TD3+BC involves explicit policy constraint (BC loss), UWAC relies on uncertainty estimation for training, and IQL learns without querying OOD samples.

We conduct experiments on MuJoCo locomotion tasks, which are made up of five types of datasets (random, medium, medium-replay, medium-expert, and expert), yielding a total of 15 datasets. We use the most recently released "-v2" datasets for performance evaluation. The results of BC and SAC are acquired by using our implemented code. The results of CQL and UWAC are obtained by running their official codes, because the reported scores in their papers are not obtained on MuJoCo "-v2" datasets. We take the results of TD3+BC from its original paper (Table 7 in [13]). Since the IQL paper does not report its performance on MuJoCo *random* and *expert* datasets, we run IQL using the official codebase on them and take the results on medium, medium-replay, medium-expert datasets from its original paper directly. All methods are run for 1M gradient steps.

Table 1: Normalized average score comparison of MCQ against baseline methods on D4RL benchmarks over the final 10 evaluations. 0 corresponds to a random policy and 100 corresponds to an expert policy. The experiments are run on MuJoCo "-v2" datasets over 4 random seeds. r = random, m = medium, m-r = medium-replay, m-e = medium-expert, e = expert. We **bold** the highest mean.

| Task Name | BC | SAC | CQL | UWAC | TD3+BC | IQL | MCQ (ours) |
|---|---|---|---|---|---|---|---|
| halfcheetah-r | 2.2±0.0 | **29.7**±1.4 | 17.5±1.5 | 2.3±0.0 | 11.0±1.1 | 13.1±1.3 | 28.5±0.6 |
| hopper-r | 3.7±0.6 | 9.9±1.5 | 7.9±0.4 | 2.7±0.3 | 8.5±0.6 | 7.9±0.2 | **31.8**±0.5 |
| walker2d-r | 1.3±0.1 | 0.9±0.8 | 5.1±1.3 | 2.0±0.4 | 1.6±1.7 | 5.4±1.2 | **17.0**±3.0 |
| halfcheetah-m | 43.2±0.6 | 55.2±27.8 | 47.0±0.5 | 42.2±0.4 | 48.3±0.3 | 47.4±0.2 | **64.3**±0.2 |
| hopper-m | 54.1±3.8 | 0.8±0.0 | 53.0±28.5 | 50.9±4.4 | 59.3±4.2 | 66.2±5.7 | **78.4**±4.3 |
| walker2d-m | 70.9±11.0 | -0.3±0.2 | 73.3±17.7 | 75.4±3.0 | 83.7±2.1 | 78.3±8.7 | **91.0**±0.4 |
| halfcheetah-m-r | 37.6±2.1 | 0.8±1.0 | 45.5±0.7 | 35.9±3.7 | 44.6±0.5 | 44.2±1.2 | **56.8**±0.6 |
| hopper-m-r | 16.6±4.8 | 7.4±0.5 | 88.7±12.9 | 25.3±1.7 | 60.9±18.8 | 94.7±8.6 | **101.6**±0.8 |
| walker2d-m-r | 20.3±9.8 | -0.4±0.3 | 81.8±2.7 | 23.6±6.9 | 81.8±5.5 | 73.8±7.1 | **91.3**±5.7 |
| halfcheetah-m-e | 44.0±1.6 | 28.4±19.4 | 75.6±25.7 | 42.7±0.3 | **90.7**±4.3 | 86.7±5.3 | 87.5±1.3 |
| hopper-m-e | 53.9±4.7 | 0.7±0.0 | 105.6±12.9 | 44.9±8.1 | 98.0±9.4 | 91.5±14.3 | **111.2**±0.1 |
| walker2d-m-e | 90.1±13.2 | 1.9±3.9 | 107.9±1.6 | 96.5±9.1 | 110.1±0.5 | 109.6±1.0 | **114.2**±0.7 |
| Average Above | 36.5 | 11.3 | 59.1 | 37.0 | 58.2 | 59.9 | **72.8** |
| halfcheetah-e | 91.8±1.5 | -0.8±1.8 | 96.3±1.3 | 92.9±0.6 | **96.7**±1.1 | 95.0±0.5 | 96.2±0.4 |
| hopper-e | 107.7±0.7 | 0.7±0.0 | 96.5±28.0 | 110.5±0.5 | 107.8±7 | 109.4±0.5 | **111.4**±0.4 |
| walker2d-e | 106.7±0.2 | 0.7±0.3 | 108.5±0.5 | 108.4±0.4 | **110.2**±0.3 | 109.9±1.2 | 107.2±1.1 |
| Total Average | 49.6 | 9.0 | 67.3 | 50.4 | 67.6 | 68.9 | **79.2** |

In our experiments, we set the number of sampled actions $N = 10$ by default and tune the weighting coefficient $\lambda$. We report the $\lambda$ used for all tasks in Appendix C, along with details on the experiments and implementation. We summarize the normalized average score comparison of MCQ against recent baselines in Table 1. Unsurprisingly, we observe that MCQ behaves better than BC on all of the tasks, which is consistent with our theoretical analysis in Proposition 2 and 3. MCQ also significantly outperforms the base SAC algorithm. Prior offline RL methods struggle for good performance on non-expert datasets like random and medium-replay, while MCQ surpasses them with a remarkable margin on many non-expert datasets. We attribute the less satisfying performance of prior offline RL methods to their *strict conservatism*, which restricts their generalization beyond the support of the dataset and leads to limited performance. The results, therefore, validate our claim that milder pessimism is more we need for offline learning. Furthermore, MCQ is also competitive to baselines on expert datasets. MCQ achieves the best performance on 11 out of 15 datasets, yielding a total average score of **72.8** on non-expert datasets, and an average score of **79.2** on all 15 datasets. Whereas the second best method, IQL, has an average score of 59.9 on non-expert datasets and a total average score of 68.9 across all tasks.

## 4.2 Parameter Study

In this subsection, we conduct a detailed parameter study on MCQ. MCQ generally contains two hyperparameters, weighting coefficient $\lambda$ and number of sampled actions $N$. To demonstrate the parameter sensitivity of MCQ, we choose two datasets from MuJoCo locomotion tasks and conduct

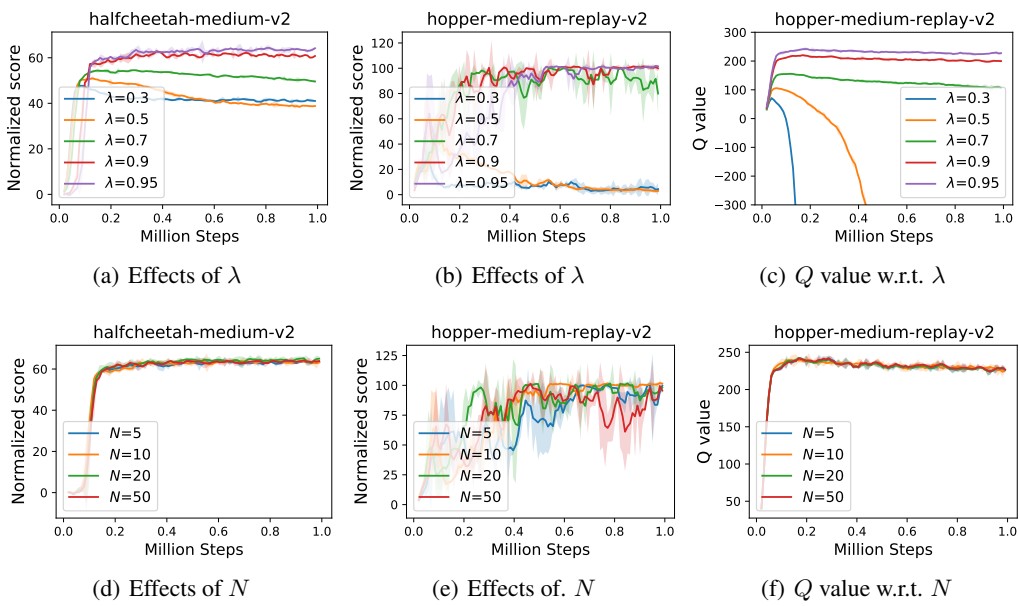

(a) Effects of $\lambda$       (b) Effects of $\lambda$       (c) $Q$ value w.r.t. $\lambda$

(d) Effects of $N$       (e) Effects of. $N$       (f) $Q$ value w.r.t. $N$

Figure 2: Parameter study and $Q$ function estimation on halfcheetah-medium-v2 and hopper-medium-replay-v2. The shaded region captures the standard deviation.

experiments on them, halfcheetah-medium-v2, and hopper-medium-replay-v2. The experiments are run for 1M gradient steps over 4 different random seeds.

**Weighting coefficient** $\lambda$**.** The weighting coefficient $\lambda$ is a critical hyperparameter for MCQ, which directly controls the balance between in-distribution actions training and OOD actions training. If we set $\lambda = 1$, then MCQ degenerates into the base SAC algorithm. If $\lambda$ leans towards 0, the critics will be overwhelmed by OOD actions. Intuitively, one ought not to use small $\lambda$, because more weights are desired for standard Bellman error such that in-distribution state-action pairs can be well-trained. We observe significant performance drop with smaller $\lambda$ in Figure 2(a) and 2(b). Also, we find that choosing $0.7 \leq \lambda < 1$ generally induces good performance.

**Number of sampled actions** $N$**.** $N$ works as a regularizer to control the potential extrapolation error. In case the behavior policy $\mu$ is known, we require $N$ to be as large as possible to better estimate the maximum $Q$ value. While in practice, we leverage the CVAE to approximate $\mu$, from which OOD actions can be sampled. $N$ then plays a role to balance pessimism and generalization. To see the influence of $N$, we fix $\lambda = 0.95$ for the two datasets. Experimental results in Figure 2(d) and 2(e) indicate that MCQ is insensitive to $N$ for a wide range of $N$. We therefore set $N = 10$ by default.

**Value estimation.** We present the $Q$ value estimates with respect to (w.r.t.) $\lambda$ and $N$ in Figure 2(c) and 2(f). The $Q$ estimation is calculated via $\mathbb{E}_{i=1,2}\mathbb{E}_{(s,a)\sim\mathcal{D}}[Q_{\theta_i}(s,a)]$. The results illustrate that (1) smaller $\lambda$ will incur severe underestimation issue (as depicted by Figure 2(c), $Q$ values collapse with $\lambda = 0.5$ or $\lambda = 0.3$); (2) no overestimation is observed, even with a large $\lambda = 0.95$, which validates the theoretical result in Proposition 5; (3) the $Q$ estimates resemble each other under different $N$. We conclude that MCQ ensures a stable and good value estimation with a proper $\lambda$.

### 4.3 Offline-to-online Fine-tuning

We examine the offline-to-online fine-tuning capability of MCQ against some prior strong offline RL baselines, CQL [35], TD3+BC [13], IQL [32]. We additionally compare against AWAC [45], which is designed intrinsically for offline-to-online adaptation. We conduct experiments on MuJoCo *random* and *medium-replay* datasets. It is challenging to train on these datasets for both offline and offline-to-online fine-tuning as they are non-expert, or even contain many bad transitions. We first train baselines and MCQ for 1M gradient steps offline and then perform online fine-tuning for another 100K gradient steps. Note that IQL paper [32] adopts 1M steps for online fine-tuning. However, we argue that 1M steps of online interactions are even enough to train off-policy online RL algorithms

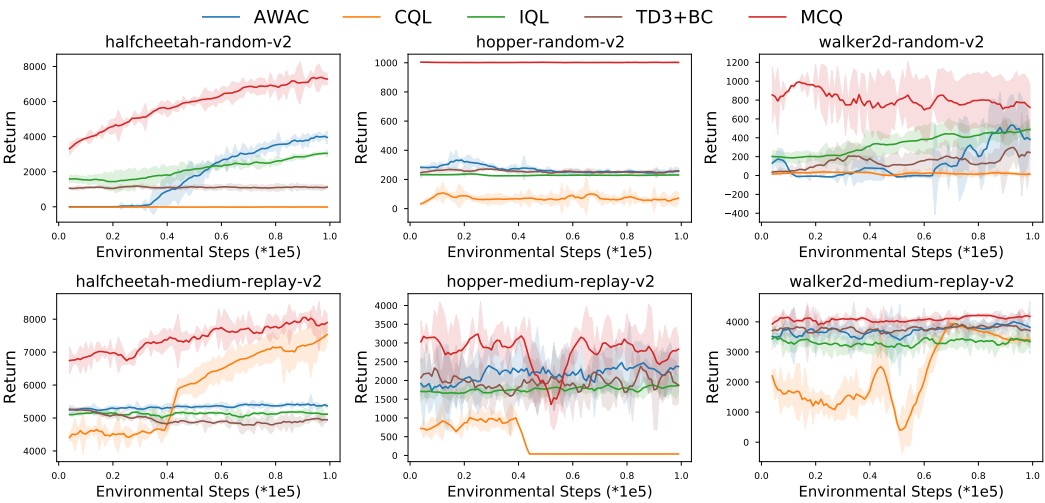

Figure 3: Offline-to-online fine-tuning results on 6 D4RL MuJoCo locomotion tasks.

from scratch to perform very well. We thus believe 100K steps is more reasonable for the online interaction. All methods are run over 4 random seeds. The results are shown in Figure 3, where the shaded region denotes the standard deviation. As expected, we observe that MCQ consistently outperforms prior offline RL methods as well as AWAC on all of the datasets, often surpassing all of them with a large margin. The mild pessimism of MCQ makes it adapt faster, or keep the offline good performance during online interactions. Other prior offline RL methods, unfortunately, fail in achieving satisfying performance during online interaction due to strict conservatism and lack of generalization ability.

## 5 Related Work

**Model-free offline RL.** Prior model-free offline RL methods are typically designed to restrict the learned policy from producing OOD actions. They usually achieve this by leveraging importance sampling [47, 54, 39, 44, 16], incorporating explicit policy constraints [34, 58, 17, 13, 11], learning latent actions [67, 3], penalizing learned value functions such that low values are assigned to unseen actions [35, 33, 41], using adaptive methods [18], and uncertainty quantification [59, 65, 4]. Another line of the methods, instead, resorts to learning without querying OOD actions [56, 8, 32]. By doing so, they constrain the learning process within the support of the dataset. Nevertheless, existing methods may induce unnecessarily over-pessimistic value functions, and their performance is largely confined by how well the behavior policy is [45, 38, 4]. That partly explains why these methods are not satisfiable when trained on non-expert datasets (e.g., random datasets). MCQ keeps milder conservatism and better generalization ability as OOD actions are *actively* trained with proper targets.

**Model-based offline RL.** Model-based offline RL methods, in contrast, learn the dynamics model in a supervised manner, and leverage the learned dynamics for policy optimization. Advances in this field include uncertainty quantification [46, 64, 27, 10], learning conservative value functions [63], representation learning [37, 48], constraining the learned policy with a behavior cloning loss [42], and sequential modelling [7, 23, 43]. However, there is no guarantee that the trained dynamics models are reliable, e.g., poor transitions can be generated, especially in complex high-dimensional environments [22]. Meanwhile, training dynamics models raises extra computation costs.

**Offline-to-online RL.** There are some efforts on accelerating online interactions with the aid of offline logged data, which is also referred to as learning from demonstration [21, 26, 49]. Offline-to-online RL, instead, aims at enhancing the well-trained offline policy via online interactions. To ensure a fast adaptation and stable policy improvement, many techniques are adopted, such as model ensemble [38], explicit policy constraints [45, 66]. Offline-to-online fine-tuning will be difficult if the trained value function or policy is overly pessimistic, which may lead to a suboptimal policy.

# 6 Conclusion

In this paper, we propose Mildly Conservative $Q$-learning (MCQ) to alleviate the over pessimism in existing offline RL methods. MCQ actively train OOD actions by constructing them proper pseudo target values following the guidance of the practical Mildly Conservative Bellman (MCB) operator. We theoretically illustrate that the policy induced by the MCB operator behaves at least as well as the behavior policy, and no erroneous overestimation will occur for the practical MCB operator. Furthermore, we extensively compare MCQ against recent strong baselines on MuJoCo locomotion tasks. Experimental results show that MCQ surpasses these baselines with a large margin on many non-expert datasets, and is also competitive with baselines on expert datasets. Moreover, we demonstrate the superior generalization capability of MCQ when transferring from offline to online. These altogether reveal that mild conservatism is critical for offline learning. We hope this work can promote the offline RL towards mild pessimism, and bring new insights into the community.

One drawback of our current algorithm lies in the need of tuning the weighting coefficient $\lambda$. However, we empirically find that $0.7 \leq \lambda < 1$ can usually induce satisfying performance. We leave the automatic tuning of $\lambda$ as future work.

## Acknowledgments and Disclosure of Funding

This work was supported in part by the Science and Technology Innovation 2030-Key Project under Grant 2021ZD0201404, in part by the NSF China under Grant 61872009. The authors would like to thank the anonymous reviewers for their valuable comments and advice.

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
