# A Missing Proofs

We recall the definition of the MCB operator below.

**Definition 3.** *The Mildly Conservative Bellman (MCB) operator is defined as*

$$\mathcal{T}_{\text{MCB}}Q(s,a) = (\mathcal{T}_1\mathcal{T}_2)Q(s,a), \tag{15}$$

*where*

$$\mathcal{T}_1 Q(s,a) = \begin{cases} Q(s,a), & \mu(a|s) > 0. \\ \max_{a' \sim \text{Support}(\mu(\cdot|s))} Q(s,a') - \delta, & else. \end{cases} \tag{16}$$

$$\mathcal{T}_2 Q(s,a) = \begin{cases} r(s,a) + \gamma\mathbb{E}_{s'}\left[\max_{a' \in \mathcal{A}} Q(s',a')\right], & \mu(a|s) > 0, \\ Q(s,a), & else. \end{cases} \tag{17}$$

**Proposition 6.** *In the support region of the behavior policy, i.e., $\text{Support}(\mu)$, the MCB operator is a $\gamma$-contraction operator in the $\mathcal{L}_\infty$ norm, and any initial $Q$ function can converge to a unique fixed point by repeatedly applying $\mathcal{T}_{\text{MCB}}$.*

*Proof.* Let $Q_1$ and $Q_2$ be two arbitrary $Q$ functions. Since $a \in \text{Support}(\mu(\cdot|s))$, then if $a' \in \text{Support}(\mu(\cdot|s'))$, we have

$$\|\mathcal{T}_{\text{MCB}}Q_1 - \mathcal{T}_{\text{MCB}}Q_2\|_\infty = \|\mathcal{T}_2 Q_1 - \mathcal{T}_2 Q_2\|_\infty$$

$$= \max_{s,a}\left|\left(r(s,a) + \gamma\mathbb{E}_{s'}\left[\max_{a'\in\mathcal{A}} Q_1(s',a')\right]\right) - \left(r(s,a) + \gamma\mathbb{E}_{s'}\left[\max_{a'\in\mathcal{A}} Q_2(s',a')\right]\right)\right|$$

$$= \gamma\max_{s,a}\left|\mathbb{E}_{s'}\left[\max_{a'\sim\mathcal{A}} Q_1(s',a') - \max_{a'\sim\mathcal{A}} Q_2(s',a')\right]\right|$$

$$\leq \gamma\max_{s,a}\mathbb{E}_{s'}\left|\max_{a'\sim\mathcal{A}} Q_1(s',a') - \max_{a'\sim\mathcal{A}} Q_2(s',a')\right|$$

$$\leq \gamma\max_{s,a}\|Q_1 - Q_2\|_\infty$$

$$= \gamma\|Q_1 - Q_2\|_\infty.$$

While if $a' \notin \text{Support}(\mu(\cdot|s'))$, then $\mathcal{T}_1$ in the MCB operator will work and assign OOD actions pseudo target values. Similarly, if $a' \notin \text{Support}(\mu(\cdot|s'))$, we have

$$\|\mathcal{T}_{\text{MCB}}Q_1 - \mathcal{T}_{\text{MCB}}Q_2\|_\infty$$
$$= \|\mathcal{T}_1\mathcal{T}_2 Q_1 - \mathcal{T}_1\mathcal{T}_2 Q_2\|_\infty$$
$$= \max_{s,a}\left|\left(r(s,a) + \gamma\mathbb{E}_{s'}\left[\max_{a'\in\text{Support}(\mu(\cdot|s'))} Q_1(s',a') - \delta\right]\right) - \left(r(s,a) + \gamma\mathbb{E}_{s'}\left[\max_{a'\in\text{Support}(\mu(\cdot|s'))} Q_2(s',a') - \delta\right]\right)\right|$$
$$= \gamma\max_{s,a}\left|\mathbb{E}_{s'}\left[\max_{a'\in\text{Support}(\mu(\cdot|s'))} Q_1(s',a') - \max_{a'\in\text{Support}(\mu(\cdot|s'))} Q_2(s',a')\right]\right|$$
$$\leq \gamma\max_{s,a}\mathbb{E}_{s'}\left|\max_{a'\in\text{Support}(\mu(\cdot|s'))} Q_1(s',a') - \max_{a'\in\text{Support}(\mu(\cdot|s'))} Q_2(s',a')\right|$$
$$\leq \gamma\max_{s,a}\|Q_1 - Q_2\|_\infty$$
$$= \gamma\|Q_1 - Q_2\|_\infty.$$

Combining the results together, we conclude that the MCB operator is a $\gamma$-contraction in the support region of the behavior policy, which naturally leads to the conclusion that any initial $Q$ function will converge to a unique fixed point by repeatedly applying $\mathcal{T}_{\text{MCB}}$. $\square$

In order to show Proposition 7, we first have the following observations.

**Observation 1.** *In the support region of $\mu$, $Q_\mu$ is the fixed point of the Bellman operator $\mathcal{T}^\mu Q(s,a) := r(s,a) + \gamma\mathbb{E}_{s'}\mathbb{E}_{a'\sim\mu(\cdot|s')}[Q(s',a')]$.*

*Proof.* It is well known that the Bellman operator is a $\gamma$-contraction in the $\mathcal{L}_\infty$ norm, and hence for some arbitrarily initialized $Q$ function, it is guaranteed to converge to a unique fixed point by repeatedly applying $\mathcal{T}^\mu$. Hence, $Q_\mu$ is the $Q$ function of the behavior policy $\mu$ in $\text{Support}(\mu)$. $\square$

**Observation 2.** *In the support region of $\mu$, $Q_{\mu^*}$ is the fixed point of the Bellman optimal operator $\mathcal{T}Q(s,a) := r(s,a) + \gamma\mathbb{E}_{s'}\left[\max_{a'\sim\mu(\cdot|s')} Q(s',a')\right]$.*

*Proof.* The proof is similar to Observation 1. Note that the resulting fixed point $Q_{\mu^*}$ is the $Q$ function of the optimal policy in the support region of the behavior policy. In offline RL setting, we only have a static logged dataset, then $Q_{\mu^*}$ is the $Q$ function of the optimal batch-constraint policy. $\square$

**Proposition 7** (Behave at least as well as behavior policy). *Denote $Q_{\mathrm{MCB}}$ as the unique fixed point acquired by the MCB operator, then in $\mathrm{Support}(\mu)$ we have: $Q_\mu \leq Q_{\mathrm{MCB}} \leq Q_{\mu^*}$, where $Q_\mu$ is the $Q$ function of the behavior policy and $Q_{\mu^*}$ is the $Q$ function of the optimal policy in the batch.*

*Proof.* Denote $\mathcal{T}Q(s,a)$ as the Bellman optimal operator and $\mathcal{T}^\mu Q(s,a)$ as the Bellman operator. Since $Q_{\mathrm{MCB}}$ is the fixed point of the MCB operator, then if $a' \in \mathrm{Support}(\mu)$, we have

$$
\begin{aligned}
Q_{\mathrm{MCB}} = \mathcal{T}_{\mathrm{MCB}}Q(s,a) &= \mathcal{T}_2 Q(s,a) \\
&= r(s,a) + \gamma \mathbb{E}_{s'} \left[ \max_{a' \in \mathrm{Support}(\mu(\cdot|s'))} Q(s',a') \right] \\
&= \mathcal{T}Q(s,a) = Q_{\mu^*}(s,a).
\end{aligned}
\tag{18}
$$

Furthermore, we have

$$
\begin{aligned}
Q_{\mathrm{MCB}} = \mathcal{T}_{\mathrm{MCB}}Q(s,a) &= \mathcal{T}_2 Q(s,a) \\
&= r(s,a) + \gamma \mathbb{E}_{s'} \left[ \max_{a' \in \mathrm{Support}(\mu(\cdot|s'))} Q(s',a') \right] \\
&\geq r(s,a) + \gamma \mathbb{E}_{s'} \mathbb{E}_{a' \in \mathrm{Support}(\mu(\cdot|s'))} [Q(s',a')] \\
&= \mathcal{T}^\mu Q(s,a) = Q_\mu(s,a).
\end{aligned}
\tag{19}
$$

If $a' \notin \mathrm{Support}(\mu)$, we have

$$
\begin{aligned}
Q_{\mathrm{MCB}} = \mathcal{T}_{\mathrm{MCB}}Q(s,a) &= \mathcal{T}_1 \mathcal{T}_2 Q(s,a) \\
&= r(s,a) + \gamma \mathbb{E}_{s'} \left[ \max_{a' \in \mathrm{Support}(\mu(\cdot|s'))} Q(s',a') - \delta \right] \\
&< \mathcal{T}Q(s,a) = Q_{\mu^*}(s,a).
\end{aligned}
\tag{20}
$$

Moreover, by setting $0 < \delta \leq \mathbb{E}_{s'} \left[ \max_{a' \in \mathrm{Support}(\mu(\cdot|s'))} Q(s',a') - \mathbb{E}_{a' \sim \mathrm{Support}(\mu(\cdot|s'))}[Q(s',a')] \right]$, we have

$$
\begin{aligned}
Q_{\mathrm{MCB}} = \mathcal{T}_{\mathrm{MCB}}Q(s,a) &= \mathcal{T}_1 \mathcal{T}_2 Q(s,a) \\
&= r(s,a) + \gamma \mathbb{E}_{s'} \left[ \max_{a' \in \mathrm{Support}(\mu(\cdot|s'))} Q(s',a') - \delta \right] \\
&\geq r(s,a) + \gamma \mathbb{E}_{s'} \mathbb{E}_{a' \in \mathrm{Support}(\mu(\cdot|s'))} [Q(s',a')] \\
&= \mathcal{T}^\mu Q(s,a) = Q_\mu(s,a).
\end{aligned}
\tag{21}
$$

Then by combining Eq. (18) and Eq. (20), the RHS holds. By combining Eq. (19) and Eq. (21), the LHS holds, which concludes the proof. $\square$

**Remark:** Note that $\delta$ is a small positive number that can be arbitrarily small. As is explained in the main text (see Section 3.1), we subtract a small positive numer such that no OOD actions will be executed, considering the fact that the action is taken by following $\arg\max_{a \in \mathcal{A}} Q(s,a)$. We actively assign pseudo target values for the OOD actions, and if the allocated value is identical to the optimal $Q$ value, actions that lie outside of the span of the behavior policy $\mu$ may be chosen and executed, which may be unsafe. By subtracting a small $\delta$, we ensure that the OOD action training will not affect the learning for the optimal batch-constraint policy.

**Proposition 8** (Milder Pessimism). *Suppose there exists an explicit policy constraint offline reinforcement learning algorithm such that the KL-divergence of the learned policy $\pi_p(\cdot|s)$ and the behavior policy $\mu(\cdot|s)$ is optimized to guarantee $\max\left(\mathrm{KL}(\mu, \pi_p), \mathrm{KL}(\pi_p, \mu)\right) \leq \epsilon, \forall s$. Denote $\epsilon_\mu^{\pi_p} = \max_s |\mathbb{E}_{a \sim \pi_p} A^\mu(s,a)|$, where $A^\mu(s,a)$ is the advantage function. Then*

$$
J(\pi_p) \geq J(\mu) - \frac{\sqrt{2}\gamma\epsilon_\mu^{\pi_p}}{(1-\gamma)^2}\sqrt{\epsilon},
\tag{22}
$$

*while for the policy $\pi_{\mathrm{MCB}}$ learned by applying the MCB operator, we have*

$$
J(\pi_{\mathrm{MCB}}) \geq J(\mu).
\tag{23}
$$

*Proof.* By utilizing the performance difference lemma (Lemma 6.1 in [25]), the return difference between two policies $\pi'$ and $\pi$ gives:

$$
J(\pi') - J(\pi) = \frac{1}{1-\gamma} \sum_s d_{\pi'}(s) \sum_a \left[ \pi'(a|s) A^\pi(s,a) \right],
\tag{24}
$$

where $d_{\pi'}(s)$ is the probability distribution underlying the states present in the static dataset $\mathcal{D}$. Furthermore, based on the Corollary 1 in [1], we have:

$$J(\pi') - J(\pi) \geq \frac{1}{1-\gamma} \sum_s d_\pi(s) \sum_a \left[ \pi'(a|s) A^\pi(s,a) \right] - \frac{2\gamma \epsilon_\pi^{\pi'}}{(1-\gamma)^2} D_{\mathrm{TV}}^{d_\pi}(\pi', \pi), \tag{25}$$

where $\epsilon_\pi^{\pi'} = \max_s |\mathbb{E}_{a \sim \pi'} A^\pi(s,a)|$ and $D_{\mathrm{TV}}^{d_\pi}(p,q)$ denotes the total variation between two distributions $p(\cdot)$ and $q(\cdot)$ over the distribution $d_\pi$, i.e., $D_{\mathrm{TV}}^{d_\pi}(p,q) = \frac{1}{2} \sum_{d_\pi} \sum_x |p(x) - q(x)|$. We use $D_{\mathrm{TV}}(p,q)$ to denote the total variation between two distributions $p(\cdot)$ and $q(\cdot)$, $D_{\mathrm{TV}}(p,q) = \frac{1}{2} \sum_x |p(x) - q(x)|$.

Following similar procedure as Proposition 2 in [56], we have

$$
\begin{aligned}
J(\pi_p) - J(\mu) &\geq \frac{1}{1-\gamma} \sum_s d_\mu(s) \sum_a \pi_p(a|s) A^\mu(s,a) - \frac{2\gamma \epsilon_\mu^{\pi_p}}{(1-\gamma)^2} D_{\mathrm{TV}}^{d_\mu}(\pi_p, \mu) \\
&\geq 0 - \frac{2\gamma \epsilon_\mu^{\pi_p}}{(1-\gamma)^2} D_{\mathrm{TV}}^{d_\mu}(\pi_p, \mu) \\
&= -\frac{2\gamma \epsilon_\mu^{\pi_p}}{(1-\gamma)^2} D_{\mathrm{TV}}^{d_\mu}(\pi_p, \mu),
\end{aligned}
\tag{26}
$$

Then, we have

$$
\begin{aligned}
J(\pi_p) - J(\mu) &\geq -\frac{2\gamma \epsilon_\mu^{\pi_p}}{(1-\gamma)^2} D_{\mathrm{TV}}^{d_\mu}(\pi_p, \mu) \\
&= -\frac{2\gamma \epsilon_\mu^{\pi_p}}{(1-\gamma)^2} \sum_s d_\mu(s) \left[ D_{\mathrm{TV}}(\pi_p(\cdot|s), \mu(\cdot|s)) \right] \\
&\geq -\frac{\sqrt{2}\gamma \epsilon_\mu^{\pi_p}}{(1-\gamma)^2} \sum_s d_\mu(s) \min\left( \sqrt{\mathrm{KL}(\mu(\cdot|s), \pi_p(\cdot|s))}, \sqrt{\mathrm{KL}(\pi_p(\cdot|s), \mu(\cdot|s))} \right) \\
&\geq -\frac{\sqrt{2}\gamma \epsilon_\mu^{\pi_p}}{(1-\gamma)^2} \sum_s d_\mu(s) \max\left( \sqrt{\mathrm{KL}(\mu(\cdot|s), \pi_p(\cdot|s))}, \sqrt{\mathrm{KL}(\pi_p(\cdot|s), \mu(\cdot|s))} \right) \\
&\geq -\frac{\sqrt{2}\gamma \epsilon_\mu^{\pi_p}}{(1-\gamma)^2} \sqrt{\epsilon}.
\end{aligned}
\tag{27}
$$

The third line above uses the Pinsker-Csiszatr inequality [9], and the fourth line holds due to the fact that $\max(a,b) \geq \min(a,b)$, and the final line holds due to the assumption we make, i.e., the KL-divergence between the learned policy and the behavior policy is optimized to guarantee $\max\left(\mathrm{KL}(\mu, \pi_p), \mathrm{KL}(\pi_p, \mu)\right) \leq \epsilon$, $\forall s$. We hence conclude that the explicit policy constraint RL algorithms induce a safe policy improvement, i.e., $J(\pi_p) \geq J(\mu) - \mathcal{O}\left(\frac{1}{(1-\gamma)^2}\right)$.

For the policy induced by the MCB operator, however, we have

$$
\begin{aligned}
J(\pi_{\mathrm{MCB}}) &= \mathbb{E}_{s \sim \mathcal{D}} \left[ V_{\pi_{\mathrm{MCB}}}(s) \right] \\
&= \mathbb{E}_{s \sim \mathcal{D}} \mathbb{E}_{a \sim \pi_{\mathrm{MCB}}} \left[ Q_{\mathrm{MCB}}(s,a) \right] \\
&= \mathbb{E}_{s \sim \mathcal{D}} \mathbb{E}_{a \sim \mathrm{Support}(\mu(\cdot|s))} \left[ Q_{\mathrm{MCB}}(s,a) \right] \\
&\geq \mathbb{E}_{s \sim \mathcal{D}} \mathbb{E}_{a \sim \mathrm{Support}(\mu(\cdot|s))} \left[ Q_\mu(s,a) \right] \quad \text{(By using Proposition 7)} \\
&= J(\mu).
\end{aligned}
\tag{28}
$$

The third line holds as the policy induced by the MCB operator will not execute actions that lie outside of the support region of behavior policy. That allows us to conclude the proof. $\qquad \square$

**Remark:** Note that in Theorem 3.6 of [35], CQL also guarantees a safe policy improvement, i.e., $J(\pi_{\mathrm{CQL}}) \geq J(\mu) - \mathcal{O}\left(\frac{1}{(1-\gamma)^2}\right)$, which is consistent with the lower bound of the explicit policy constraint methods. These methods are often overly pessimistic, while the policy learned by applying the MCB operator can improve the policy with a tighter lower bound.

**Proposition 9.** *Proposition 6 still holds for the practical MCB operator.*

*Proof.* The practical MCB operator only modifies $\mathcal{T}_1$, i.e., the target value for the OOD actions. We only discuss $a' \notin \mathrm{Support}(\mu(\cdot|s'))$ here because for $a' \in \mathrm{Support}(\mu(\cdot|s'))$, the proof is identical as Proposition 6. For

$a' \notin \text{Support}(\mu(\cdot|s'))$, let $Q_1$ and $Q_2$ be two arbitrary $Q$ functions, then

$$\|\hat{\mathcal{T}}_{\text{MCB}}Q_1 - \hat{\mathcal{T}}_{\text{MCB}}Q_2\|_\infty$$

$$= \|\hat{\mathcal{T}}_1\mathcal{T}_2Q_1 - \hat{\mathcal{T}}_1\mathcal{T}_2Q_2\|_\infty$$

$$= \max_{s,a}\left|\left(r(s,a) + \gamma\mathbb{E}_{s'\sim\mathcal{D}}\mathbb{E}_{\{a_i'\}^N\sim\text{Support}(\hat{\mu})}\left[\max_{a'\in\{a_i'\}^N}Q_1(s',a')\right]\right)\right.$$

$$\left. - \left(r(s,a) + \gamma\mathbb{E}_{s'\sim\mathcal{D}}\mathbb{E}_{\{a_i'\}^N\sim\text{Support}(\hat{\mu})}\left[\max_{a'\in\{a_i'\}^N}Q_2(s',a')\right]\right)\right|$$

$$= \gamma\max_{s,a}\left|\mathbb{E}_{s'\sim\mathcal{D}}\mathbb{E}_{\{a_i'\}^N\sim\text{Support}(\hat{\mu})}\left[\max_{a'\in\{a_i'\}^N}Q_1(s',a') - \max_{a'\in\{a_i'\}^N}Q_2(s',a')\right]\right|$$

$$\leq \gamma\max_{s,a}\mathbb{E}_{s'\sim\mathcal{D}}\mathbb{E}_{\{a_i'\}^N\sim\text{Support}(\hat{\mu})}\left|\max_{a'\sim\{a_i'\}^N}Q_1(s',a') - \max_{a'\sim\{a_i'\}^N}Q_2(s',a')\right|$$

$$\leq \gamma\max_{s,a}\mathbb{E}_{s'\sim\mathcal{D}}\mathbb{E}_{\{a_i'\}^N}\|Q_1 - Q_2\|_\infty$$

$$= \gamma\|Q_1 - Q_2\|_\infty,$$

where we use a fact that $\max_{a'\in\{a_i'\}^N}Q_1(s',a') - \max_{a'\in\{a_i'\}^N}Q_2(s',a') \leq \|Q_1 - Q_2\|_\infty$. Its proof is quite straightforward. Let $\hat{a} = \arg\max_{\{a_i'\}^N}Q_1(s',a_i')$, then

$$\max_{a'\in\{a_i'\}^N}Q_1(s',a') - \max_{a'\in\{a_i'\}^N}Q_2(s',a') = Q_1(s',\hat{a}) - \max_{a'\in\{a_i'\}^N}Q_2(s',a')$$

$$\leq Q_1(s',\hat{a}) - Q_2(s',\hat{a}) \leq \|Q_1 - Q_2\|_\infty.$$

Therefore, we conclude that Proposition 6 still holds for the practical MCB operator. $\square$

**Proposition 10** (No erroneous overestimation will occur). *Assuming that* $\sup_s D_{\text{TV}}(\hat{\mu}(\cdot|s) \parallel \mu(\cdot|s)) \leq \epsilon < \frac{1}{2}$, *we have*

$$\mathbb{E}_{\{a_i'\}^N\sim\hat{\mu}(\cdot|s)}\left[\max_{a'\in\{a_i'\}^N}Q(s,a')\right] \leq \max_{a'\in\text{Support}(\mu(\cdot|s))}Q(s,a') + (1-(1-2\epsilon)^N)\frac{r_{\max}}{1-\gamma}.$$

*Proof.* The total variation divergence $D_{TV}(p\|q) = \frac{1}{2}\sum_x|p(x) - q(x)|$ for some distributions $p(\cdot)$ and $q(\cdot)$, then we have

$$1 > 2\epsilon \geq 2\sup_s D_{\text{TV}}(\hat{\mu}(\cdot|s) \parallel \mu(\cdot|s)) \geq \sum_a|\hat{\mu}(a|s) - \mu(a|s)|$$

$$= \sum_{a\in\text{Support}(\mu(\cdot|s))}|\hat{\mu}(a|s) - \mu(a|s)| + \sum_{a\notin\text{Support}(\mu(\cdot|s))}|\hat{\mu}(a|s) - \mu(a|s)|$$

$$\geq \sum_{a\notin\text{Support}(\mu(\cdot|s))}\hat{\mu}(a|s),$$

(29)

where we use the fact that $\mu(a|s) = 0$ if $a \notin \text{Support}(\mu(\cdot|s))$. Denote $Q_{\max}$ as the maximum value of the OOD actions, and $Q_{\max} \leq \frac{r_{\max}}{1-\gamma}$. Thus, we have

$$\mathbb{E}_{\{a_i'\}^N\sim\hat{\mu}(\cdot|s)}\left[\max_{a'\in\{a_i'\}^N}Q(s,a')\right]$$

$$\leq \mathbb{P}\left(\bigcap_i\{a_i'\in\text{Support}(\mu(\cdot|s))\}\right)\max_{a'\in\text{Support}(\mu(\cdot|s))}Q(s,a') + \mathbb{P}\left(\bigcup_i\{a_i'\notin\text{Support}(\mu(\cdot|s))\}\right)Q_{\max}$$

$$\leq \max_{a'\in\text{Support}(\mu(\cdot|s))}Q(s,a') + \left(1 - \mathbb{P}(a_1'\in\text{Support}(\mu(\cdot|s)))^N\right)\frac{r_{\max}}{1-\gamma}$$

$$\leq \max_{a'\in\text{Support}(\mu(\cdot|s))}Q(s,a') + \left(1 - (1-2\epsilon)^N\right)\frac{r_{\max}}{1-\gamma}.$$

$\square$

**Remark:** Note that we empirically fit a behavior policy $\hat{\mu}(\cdot|s)$ as we usually do not have prior knowledge about the true behavior policy $\mu$. The conclusion says that for the practical MCB operator and empirical behavior

policy $\hat{\mu}(\cdot|s)$, even if OOD actions are sampled from the $\hat{\mu}(\cdot|s)$, the extrapolation error is under the scale of $\left(1 - (1 - 2\epsilon)^N\right) \frac{r_{\max}}{1 - \gamma}$, indicating the fact that no severe overestimation will occur. If the empirical behavior policy well fits the true behavior policy, then the error term will be arbitrarily small and the value estimate will well approximate the maximum $Q$ value in the batch. We assume that the total variation divergence between $\hat{\mu}(\cdot|s)$ and $\mu$ is bounded by $\frac{1}{2}$, which is easy to satisfy in practice, e.g., fit the empirical behavior policy via supervised learning.

# B   Deterministic MCQ

As mentioned in the main text, the practical MCB operator is pluggable and can be combined with wide range of off-policy online RL methods. In this section, we present a deterministic version of MCQ by incorporating the practical MCB operator with TD3 [15].

Similarly, we train a CVAE $G_\omega$ which is made up of an encoder $E_\xi(s, a)$ and a decoder $D_\psi(s, z)$ as the generative model, which is optimized by:

$$\mathcal{L}_{\text{CVAE}} = \mathbb{E}_{(s,a)\sim\mathcal{D}, z\sim E_\xi(s,a)} \left[ (a - D_\psi(s, z))^2 + \text{KL}\left(E_\xi(s, a), \mathcal{N}(0, \mathbf{I})\right) \right]. \tag{30}$$

The objective function for the critics gives:

$$\mathcal{L}_{\text{critic}} = \lambda \mathbb{E}_{(s,a,r,s')\sim\mathcal{D}} \left[ (Q_{\theta_i}(s, a) - y)^2 \right] + (1 - \lambda) \mathbb{E}_{s^{\text{in}}\sim\mathcal{D}} \left[ (Q_{\theta_i}(s^{\text{in}}, \pi(s^{\text{in}})) - y')^2 \right], \tag{31}$$

where $\theta_i, i = 1, 2$, is the parameter of the critic network, and the target value for the in-distribution action gives:

$$y = r(s, a) + \gamma \left[ \min_{i=1,2} Q_{\theta_i'}(s', \pi_{\phi'}(s')) \right], \tag{32}$$

where $\theta_i', i = 1, 2$ is the parameter of the lagging target network, and $\phi'$ is the parameter of the actor network. For OOD actions, the target value gives:

$$y' = \min_{j=1,2} \mathbb{E}_{\{a_i'\}^N\sim\hat{\mu}} \left[ \max_{a'\sim\{a_i'\}^N} Q_{\theta_j}(s^{\text{in}}, a') \right]. \tag{33}$$

The actor is then optimized by $\max_\phi \mathbb{E}_{s\sim\mathcal{D}} [Q_{\theta_1}(s, \pi_\phi(s))]$. The full procedure of the deterministic MCQ is presented in Algorithm 2.

---

**Algorithm 2** Deterministic version of Mildly Conservative $Q$-learning (MCQ)

---

1: Initialize CVAE $G_\omega$, critic networks $Q_{\theta_1}, Q_{\theta_2}$ and actor network $\pi_\phi$ with random parameters
2: Initialize target networks $\theta_1' \leftarrow \theta_1, \theta_2' \leftarrow \theta_2, \phi' \leftarrow \phi$ and offline replay buffer $\mathcal{D}$.
3: **for** $t = 1$ to $T$ **do**
4:      Sample a mini-batch $B = \{(s, a, r, s', d)\}$ from $\mathcal{D}$, where $d$ is the done flag
5:      Train CVAE via minimizing Eq. (30)
6:      Get target value: $y = r(s, a) + \gamma \left[ \min_{i=1,2} Q_{\theta_i'}(s', \pi_{\phi'}(s')) \right]$
7:      Set $s^{\text{in}} = \{s, s'\}$ and compute the target value for the OOD actions via Eq. (33)
8:      Update critic $\theta_i$ with gradient descent via minimizing Eq. (31)
9:      Update $\phi$ by policy gradient: $\nabla_\phi J(\phi) = |B|^{-1} \sum \nabla_a Q_{\theta_1}(s, a)|_{a=\pi_\phi(s)} \nabla_\phi \pi_\phi(s)$
10:      Update target networks: $\phi' \leftarrow \tau\phi + (1 - \tau)\phi', \theta_i' \leftarrow \tau\theta_i + (1 - \tau)\theta_i', i = 1, 2$
11: **end for**

---

Next, we empirically examine the performance of the deterministic MCQ by conducting experiments on D4RL [12] MuJoCo locomotion tasks. As we want to focus on the benefits of MCQ on non-expert datasets, and considering the fact that all baselines can achieve good performance on expert datasets, we only report the performance comparison on non-expert datasets (random, medium, medium-replay, and medium-expert). The results are summarized in Table 2. For the deterministic MCQ, we adopt identical $\lambda$ and $N$ as vanilla MCQ (see Table 5) except that we use $\lambda = 0.4$ on hopper-medium-expert-v2. We observe that the deterministic MCQ has a good performance on many non-expert datasets, which is consistent with vanilla MCQ (which is built upon SAC). The deterministic MCQ has an average score of 68.4 on non-expert datasets and the vanilla MCQ has an average score of 72.8. Note that it is not suitable to compare BC against the deterministic MCQ (though deterministic MCQ still outperforms BC on all of the datasets), since the actor in our BC is not deterministic.

All these validate our claim that the practical MCB operator is pluggable and can be combined with any off-policy online RL algorithm to make it work in the offline setting. Note that it is also very interesting to report all the numerical results based on [2], which we omit here.

Table 2: Normalized average score comparison of MCQ against baseline methods on D4RL benchmarks over the final 10 evaluations. 0 corresponds to a random policy and 100 corresponds to an expert policy. The experiments are run on MuJoCo "-v2" datasets over 4 random seeds. r = random, m = medium, m-r = medium-replay, m-e = medium-expert, e = expert. MCQ (TD3) denotes the deterministic version of MCQ that built upon TD3.

| Task Name | BC | SAC | CQL | UWAC | TD3+BC | IQL | MCQ (TD3) |
|---|---|---|---|---|---|---|---|
| halfcheetah-r | 2.2±0.0 | 29.7±1.4 | 17.5±1.5 | 2.3±0.0 | 11.0±1.1 | 13.1±1.3 | 23.6±0.8 |
| hopper-r | 3.7±0.6 | 9.9±1.5 | 7.9±0.4 | 2.7±0.3 | 8.5±0.6 | 7.9±0.2 | 31.0±1.7 |
| walker2d-r | 1.3±0.1 | 0.9±0.8 | 5.1±1.3 | 2.0±0.4 | 1.6±1.7 | 5.4±1.2 | 10.3±6.8 |
| halfcheetah-m | 43.2±0.6 | 55.2±27.8 | 47.0±0.5 | 42.2±0.4 | 48.3±0.3 | 47.4±0.2 | 58.3±1.3 |
| hopper-m | 54.1±3.8 | 0.8±0.0 | 53.0±28.5 | 50.9±4.4 | 59.3±4.2 | 66.2±5.7 | 73.6±10.3 |
| walker2d-m | 70.9±11.0 | -0.3±0.2 | 73.3±17.7 | 75.4±3.0 | 83.7±2.1 | 78.3±8.7 | 88.4±1.3 |
| halfcheetah-m-r | 37.6±2.1 | 0.8±1.0 | 45.5±0.7 | 35.9±3.7 | 44.6±0.5 | 44.2±1.2 | 51.5±0.2 |
| hopper-m-r | 16.6±4.8 | 7.4±0.5 | 88.7±12.9 | 25.3±1.7 | 60.9±18.8 | 94.7±8.6 | 99.5±1.7 |
| walker2d-m-r | 20.3±9.8 | -0.4±0.3 | 81.8±2.7 | 23.6±6.9 | 81.8±5.5 | 73.8±7.1 | 83.3±1.9 |
| halfcheetah-m-e | 44.0±1.6 | 28.4±19.4 | 75.6±25.7 | 42.7±0.3 | 90.7±4.3 | 86.7±5.3 | 85.4±3.4 |
| hopper-m-e | 53.9±4.7 | 0.7±0.0 | 105.6±12.9 | 44.9±8.1 | 98.0±9.4 | 91.5±14.3 | 106.1±2.3 |
| walker2d-m-e | 90.1±13.2 | 1.9±3.9 | 107.9±1.6 | 96.5±9.1 | 110.1±0.5 | 109.6±1.0 | 110.3±0.1 |
| Average | 36.5 | 11.3 | 59.1 | 37.0 | 58.2 | 59.9 | 68.4 |

## C  Experimental Details and Parameter Setup

In this section, we first give a brief introduction to the D4RL benchmarks [12]. Then we present our implementation details and experimental details.

### C.1  D4RL Benchmarks

In the paper, we evaluate MCQ on the D4RL MuJoCo-Gym domain, which contains three environments (halfcheetah, hopper, walker2d), and five types of datasets (random, medium, medium-replay, medium-expert, expert). **Random** datasets are gathered by a random policy. **Medium** datasets contain experience from an early-stopped SAC policy. **Medium-replay** datasets are collected during the training process of the "medium" SAC policy. **Medium-expert** datasets are formed by combining the suboptimal samples and the expert samples. **Expert** datasets are made up of expert trajectories. The illustration of MuJoCo tasks is shown in Figure 4.

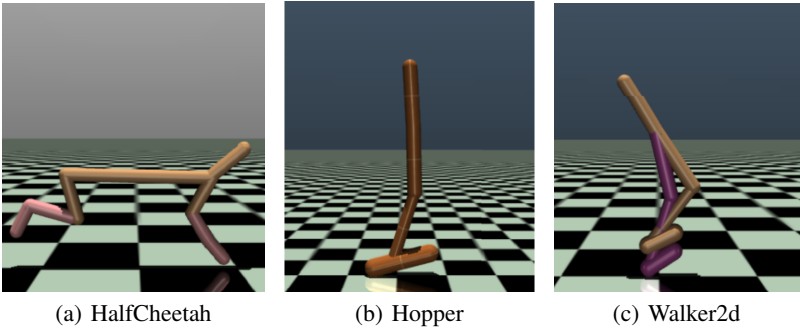

(a) HalfCheetah      (b) Hopper      (c) Walker2d

Figure 4: MuJoCo datasets. We conduct experiments on halfcheetah, hopper, and walker2d tasks.

D4RL offers a metric, normalized score, to evaluate the performance of the offline RL algorithm, which is calculated by:

$$\text{score} = \frac{\text{average return} - \text{return of the random policy}}{\text{return of the expert policy} - \text{return of the random policy}} \times 100.$$

Note that if the normalized score equals to 0, that indicates that the learned policy has a similar performance as the random policy, while 100 corresponds to an expert policy. In D4RL, different types of datasets (e.g., random,

medium, expert) share the identical reference minimum score and reference maximum score. We summarize the reference score for each environment in Table 3.

Table 3: The referenced min score and max score for the MuJoCo dataset in D4RL.

| Domain | Task Name | Reference Min Score | Reference Max Score |
|--------|-----------|---------------------|---------------------|
| MuJoCo | halfcheetah | $-280.18$ | 12135.0 |
| MuJoCo | hopper | $-20.27$ | 3234.3 |
| MuJoCo | Walker2d | 1.63 | 4592.3 |

## C.2 Implementation Details

**Baselines.** We conduct experiments on the recently released MuJoCo "-v2" datasets. We adopt behavior cloning (BC), Soft Actor-Critic (SAC) [20], and recently proposed offline RL methods, CQL [35], UWAC [59], TD3+BC [13], and IQL [32] as baseline methods. The results of CQL are obtained by running the official codebase (https://github.com/aviralkumar2907/CQL). For UWAC, we rerun it on MuJoCo "-v2" datasets using the official codebase (https://github.com/apple/ml-uwac) since the results reported in its original paper are not obtained on "-v2" datasets. For TD3+BC, we actually can reproduce its results, and thus we take the results reported in its original paper (Table 7 in the appendix) directly. For IQL, we take its results on medium, medium-replay, and medium-expert from its original paper directly. We run IQL on random and expert datasets with its official codebase (https://github.com/ikostrikov/implicit_q_learning). All methods are run over 4 different random seeds and the normalized average scores are reported in the main text.

**MCQ implementation details.** We build the practical MCB operator upon Soft Actor-Critic (SAC), and modify its critic objective function by adding an auxiliary loss such that OOD actions can be trained. For some datasets, we subtract the pseudo target value a small positive number (as in Definition 1) since we find it behaves better. We also stop overly estimated value from backpropagating for some of the datasets. We additionally train a generative model, CVAE. For each training epoch, we sample a mini-batch from the logged dataset, and train the CVAE. Next, we leverage the critic networks for estimating the $Q$-values and their corresponding target values upon in-distribution state-action pairs. Furthermore, we sample $N$ actions from the CVAE based on the current state and the next state, and sample $N$ actions from the learned policy. We then construct the pseudo target values for OOD actions. Finally, we optimize the learned policy with the standard way SAC does. We detail the default parameter setup for SAC and CVAE in Table 4.

We set the number of sampled actions $N = 10$ by default as we experimentally find that MCQ is insensitive to $N$ in a wide range. Across all of our experiments, we tune the weighting coefficient $\lambda$ across $\{0.3, 0.5, 0.6, 0.7, 0.8, 0.9, 0.95\}$ with grid search. We summarize the hyperparameters we use for running the MuJoCo datasets in Table 5, where we find that generally $0.7 \leq \lambda < 1$ is suitable for most of the datasets. For some datasets with narrow distributions, e.g., expert datasets and medium-expert datasets, a comparatively small $\lambda$ is better because actions sampled from the learned policy are more likely to lie in the OOD region in these tasks, and larger weights are expected to be assigned to OOD actions training loss. Again, as we have explained in the main text, one ought not to use too small $\lambda$ (and absolutely cannot set $\lambda = 0$) as our first priority is always training the in-distribution data in a good manner.

**Offline-to-Online details.** For offline-to-online experiments, we directly put the online samples into the offline replay buffer after offline initialization. Both offline transitions and online transitions are sampled equally during online interaction and training. The hyperparameters of all methods are kept unchanged on both offline stage and online stage. The results of baseline methods are acquired by running their official codebases, i.e., CQL (https://github.com/aviralkumar2907/CQL), TD3+BC (https://github.com/sfujim/TD3_BC), IQL (https://github.com/ikostrikov/implicit_q_learning), AWAC (https://github.com/vitchyr/rlkit). For those involve normalization over states, we normalize the online samples with the mean and standard deviation calculated based on the offline samples. All methods are run over 4 different random seeds. We choose a subset of tasks for offline-to-online fine-tuning different from IQL and AWAC to ensure that our empirical experiments on offline-to-online fine-tuning are consistent to the offline experiments on MuJoCo datasets.

# D  Comparison of MCQ against More Baselines

In this section, we extensively compare MCQ against severe recent strong offline RL methods on D4RL MuJoCo datasets, including BCQ [14], BEAR [34], MOPO [64], Decision Transformer (DT) [7], CDC [11], and PBRL [4]. BCQ aims at learning a batch-constraint policy and it also involves a generative model, i.e., CVAE. BEAR lies in the category of explicit policy constraint offline RL methods. MOPO is a well-known model-based offline RL algorithm that leverages the learned dynamics models for uncertainty quantification. Decision Transformer (DT) is also a recent model-based offline RL method, and it relies on the transformer for sequential modeling.

Table 4: Hyperparameters setup for MCQ.

| Hyperparameter | Value |
|---|---|
| SAC | |
| Actor network | $(400, 400)$ |
| Critic network | $(400, 400)$ |
| Batch size | 256 |
| Critic learning rate | $3 \times 10^{-4}$ |
| Actor learning rate | $3 \times 10^{-4}$ |
| Optimizer | Adam [28] |
| Discount factor | 0.99 |
| Reward scale | 1 |
| Maximum log std | 2 |
| Minimum log std | $-20$ |
| Use automatic entropy tuning | Yes |
| Target update rate | $5 \times 10^{-3}$ |
| CVAE | |
| Encoder hidden dimension | 750 |
| Decoder hidden dimension | 750 |
| Hidden layers | 2 |
| CVAE learning rate | $1 \times 10^{-3}$ |
| Batch size | 256 |
| Latent dimension | $2\times$ action dimension |

Table 5: Detailed hyperparameters used in MCQ, where we conduct experiments on D4RL MuJoCo-Gym "-v2" datasets.

| Task Name | weighting coefficient $\lambda$ | number of sampled actions $N$ |
|---|---|---|
| halfcheetah-random-v2 | 0.95 | 10 |
| hopper-random-v2 | 0.6 | 10 |
| walker2d-random-v2 | 0.8 | 10 |
| halfcheetah-medium-v2 | 0.95 | 10 |
| hopper-medium-v2 | 0.7 | 10 |
| walker2d-medium-v2 | 0.9 | 10 |
| halfcheetah-medium-replay-v2 | 0.95 | 10 |
| hopper-medium-replay-v2 | 0.9 | 10 |
| walker2d-medium-replay-v2 | 0.9 | 10 |
| halfcheetah-medium-expert-v2 | 0.7 | 10 |
| hopper-medium-expert-v2 | 0.5 | 10 |
| walker2d-medium-expert-v2 | 0.8 | 10 |
| halfcheetah-expert-v2 | 0.5 | 10 |
| hopper-expert-v2 | 0.5 | 10 |
| walker2d-expert-v2 | 0.3 | 10 |

CDC adds critic regularization for the critics and the explicit policy constraint for the actor. PBRL is an ensemble-based offline RL method, which also quantifies the estimation uncertainty. We conduct experiments on D4RL [12] MuJoCo locomotion datasets. Note that the results of BCQ is obtained by running its official codebase (https://github.com/sfujim/BCQ), and the results of BEAR, MOPO, and PBRL are taken directly from [4]. The results of Decision Transformer are taken from [13] (Table 7 in the appendix). Since CDC does not report standard deviation in its original paper, we omit them and directly report its mean performance. We summarize the normalized average score of each method in Table 6.

As shown, MCQ outperforms these baseline methods on most of the non-expert datasets, often by a large margin, and is competitive with them on expert datasets. Note that RBRL also leverages OOD sampling in its formulation, The main difference between MCQ and PBRL lies in the fact that PBRL adopts OOD sampling for penalizing the estimation uncertainty while we construct pseudo target values for OOD actions. Moreover, PBRL utilizes critic ensemble for uncertainty quantification while MCQ only adopts double critics. MCQ is much more computationally efficient than PBRL. One can also find that MCQ surpasses PBRL on many datasets, especially on non-expert datasets.

Table 6: Normalized average score comparison of MCQ against baseline methods on D4RL benchmarks. 0 corresponds to a random policy and 100 corresponds to an expert policy. The experiments are run on MuJoCo "-v2" datasets over 4 random seeds. r = random, m = medium, m-r = medium-replay, m-e = medium-expert, e = expert.

| Task Name | BCQ | BEAR | MOPO | DT | CDC | PBRL | MCQ (ours) |
|---|---|---|---|---|---|---|---|
| halfcheetah-r | 2.2±0.0 | 2.3±0.0 | 35.9 ±2.9 | - | 27.4 | 11.0±5.8 | 28.5 ±0.6 |
| hopper-r | 7.8±0.6 | 3.9±2.3 | 16.7±12.2 | - | 14.8 | 26.8±9.3 | 31.8 ±0.5 |
| walker2d-r | 4.9±0.1 | 12.8±10.2 | 4.2±5.7 | - | 7.2 | 8.1±4.4 | 17.0 ±3.0 |
| halfcheetah-m | 46.6±0.4 | 43.0±0.2 | 73.1 ±2.4 | 42.6±0.1 | 46.1 | 57.9±1.5 | 64.3 ±0.2 |
| hopper-m | 59.4±8.3 | 51.8±4.0 | 38.3±34.9 | 67.6±1.0 | 60.4 | 75.3 ±31.2 | 78.4 ±4.3 |
| walker2d-m | 71.8±7.2 | -0.2±0.1 | 41.2±30.8 | 74.0±1.4 | 82.1 | 89.6 ±0.7 | 91.0 ±0.4 |
| halfcheetah-m-r | 42.2±0.9 | 36.3±3.1 | 69.2 ±1.1 | 36.6±0.8 | 44.7 | 45.1±8.0 | 56.8 ±0.6 |
| hopper-m-r | 60.9±14.7 | 52.2±19.3 | 32.7±9.4 | 82.7±7.0 | 55.4 | 100.6 ±1.0 | 101.6 ±0.8 |
| walker2d-m-r | 57.0±9.6 | 7.0±7.8 | 73.7±9.4 | 66.6±3.0 | 23.0 | 77.7±14.5 | 91.3 ±5.7 |
| halfcheetah-m-e | 95.4 ±2.0 | 46.0±4.7 | 70.3±21.9 | 86.8±1.3 | 59.6 | 92.3 ±1.1 | 87.5±1.3 |
| hopper-m-e | 106.9±5.0 | 50.6±25.3 | 60.6±32.5 | 107.6±1.8 | 86.9 | 110.8 ±0.8 | 111.2 ±0.1 |
| walker2d-m-e | 107.7 ±3.8 | 22.1±44.9 | 77.4±27.9 | 108.1 ±0.2 | 70.9 | 110.1 ±0.3 | 114.2 ±0.7 |
| Average Above | 55.2 | 27.3 | 49.4 | - | 48.2 | 67.1 | 72.8 |
| halfcheetah-e | 89.9±9.6 | 92.7 ±0.6 | 81.3±21.8 | - | 82.1 | 92.4 ±1.7 | 96.2 ±0.4 |
| hopper-e | 109.0 ±4.0 | 54.6±21.0 | 62.5±29.0 | - | 102.8 | 110.5±0.4 | 111.4 ±0.4 |
| walker2d-e | 106.3 ±5.0 | 106.6 ±6.8 | 62.4±3.2 | - | 87.5 | 108.3 ±0.3 | 107.2 ±1.1 |
| Total Average | 64.5 | 38.8 | 53.3 | - | 56.7 | 74.4 | 79.2 |

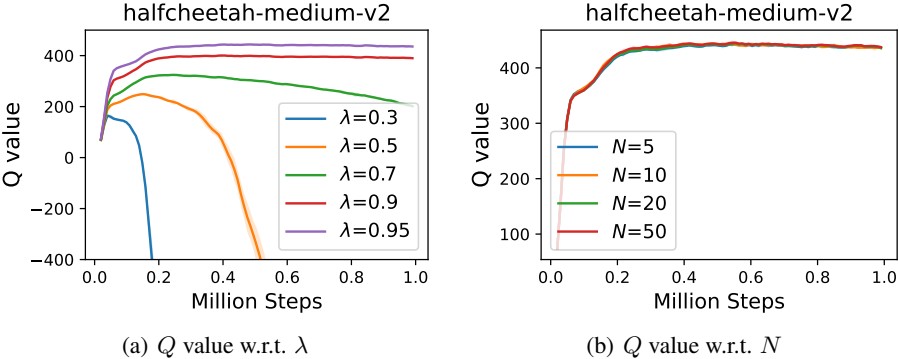

(a) $Q$ value w.r.t. $\lambda$      (b) $Q$ value w.r.t. $N$

Figure 5: Missing value estimation on halfcheetah-medium-v2.

# E    Wider Evidence on Value Estimation of MCQ

We first want to include the $Q$-value estimation with respect to the $\lambda$ and $N$ on halfcheetah-medium-v2, which we omit in the main text due to the space limit. The results are shown in Figure 5, where we can observe that the results resemble those on hopper-medium-replay-v2, e.g., the $Q$ value estimate collapses with smaller $\lambda$ and is comparatively insensitive to $N$ in a wide range.

Furthermore, we include more evidence that MCQ induces a good and stable value estimation in Figure 6 across more datasets with the parameters reported in Section C.2. We estimate the in-distribution $Q$-value by $\mathbb{E}_{s,a\sim\mathcal{D}}\mathbb{E}_{i=1,2}\left[Q_{\theta_i}(s,a)\right]$, and estimate the $Q$-value for the OOD actions via $\mathbb{E}_{s\sim\mathcal{D},a\sim\pi(\cdot|s)}\mathbb{E}_{i=1,2}\left[Q_{\theta_i}(s,a)\right]$, and the target value estimates for the in-distribution samples are give by the standard way of calculating target value in SAC. We find that on all of the datasets, MCQ has a good value estimation upon both in-distribution samples and OOD samples. The $Q$ estimation curve is quite smooth and stable on most of them. Based on these observations, we can conclude that MCQ guarantees a good value estimation and will not incur severe overestimation.

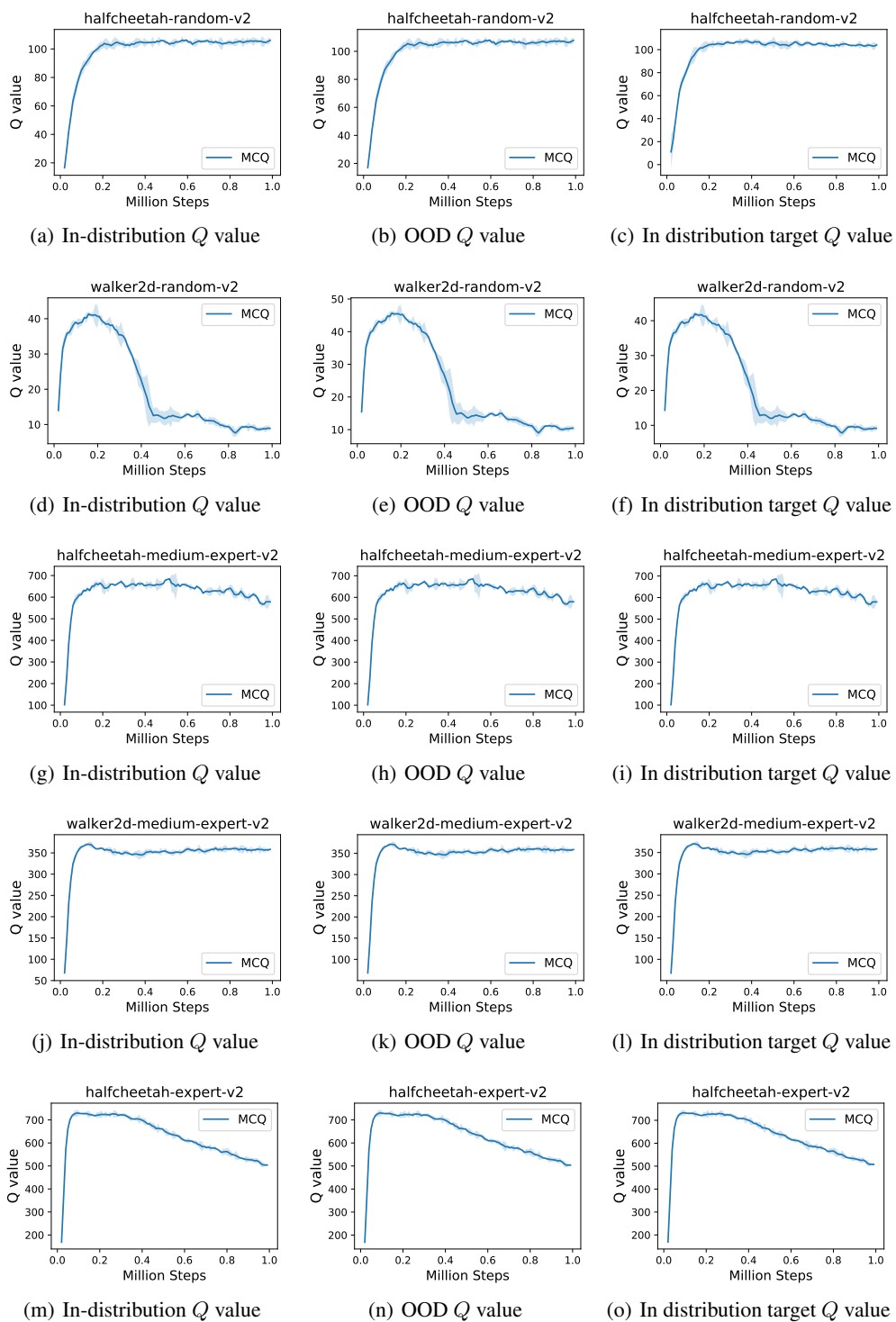

Figure 6: $Q$ value estimation of MCQ on different types of datasets, where $Q$ estimates upon in-dataset samples and OOD samples, target $Q$ value for the in-dataset samples are depicted. The shaded region captures the standard deviation.

# F Computation Cost and Compute Infrastructure

As for the computational cost, MCQ generally consumes 8 to 12 hours to train on MuJoCo tasks due to the cost of OOD sampling. In contrast, CQL takes around 12 to 15 hours. We compare MCQ against CQL on halfcheetah-medium-v2 task in terms of runtime per epoch (1K gradient steps) and GPU memory during training. We conduct the experiments on a single RTX3090 GPU. We detail the computation cost comparison of MCQ against CQL in Table 7. As shown, MCQ is computationally more efficient than CQL and can achieve better performance with lower computational costs.

Table 7: Computation cost comparison of MCQ and CQL.

| Algorithm | Runtime (s/epoch) | GPU Memory (GB) |
|:---:|:---:|:---:|
| CQL | 38.2 | 1.4 |
| MCQ | 30.3 | 1.2 |

In Table 8, we list the compute infrastructure that we use to run all of the baseline algorithms and MCQ experiments.

Table 8: Compute infrastructure.

| CPU | GPU | Memory |
|:---:|:---:|:---:|
| AMD EPYC 7452 | RTX3090×8 | 288GB |

# G Omitted Background for VAE

In this section, we give a brief introduction to the VAE, and CVAE [29, 51]. Given a dataset $\{x_i\}_{i=1}^N$, the VAE learns a generative model $p_\theta(X) = \prod_{i=1}^N p_\theta(x_i)$, where $X = \{x_i\}_{i=1}^N$ and $\theta$ is the parameter of the generative model. The goal of the VAE is to generate samples that come from the identical distribution as the samples in the dataset, which is equivalent to maximize $p_\theta(x_i)$ for all $x_i$. VAE achieves this by introducing a latent variable $z$ with a prior distribution $p(z)$ and modeling a decoder $p_\theta(X|z)$. The true posterior distribution $p_\theta(z|X)$ is approximated by training an encoder $q_\phi(z|X)$ parameterized by $\phi$. VAE then optimizes the following variational lower bound:

$$\log p_\theta(X) \geq \mathbb{E}_{z \sim q_\phi(\cdot|X)} \left[\log p_\theta(X|z)\right] - \mathrm{KL}(q_\phi(z|X), p(z)). \tag{34}$$

Note that $\log p_\theta(X|z)$ denotes the reconstruction loss, and $\mathrm{KL}(p(\cdot), q(\cdot))$ represents the KL-divergence between two probability distribution $p(\cdot)$ and $q(\cdot)$. The prior distribution $p(z)$ is usually set to be a multivariate Gaussian distribution.

Conditional variational auto-encoder (CVAE) is a variant of the vanilla VAE, which aims to model $p_\theta(X|Y)$. The corresponding variational lower bound is given below.

$$\log p_\theta(X|Y) \geq \mathbb{E}_{z \sim q_\phi(\cdot|X,Y)} \left[\log p_\theta(X|z, Y)\right] - \mathrm{KL}(q_\phi(z|X, Y), p(z|Y)). \tag{35}$$

When sampling from the CVAE, we first sample from the prior distribution $p(z|Y)$ and pass it to the decoder to get the desired sample.

# H Experimental Results on Other Datasets

To further illustrate the effectiveness of the MCQ algorithm, we conduct more experiments on 4 maze2d "-v1" datasets and 8 Adroit "-v0" datasets from D4RL benchmarks. We compare MCQ against behavior cloning (BC), BEAR [34], CQL [35], BCQ [14], TD3+BC [13], and IQL [32]. We take the results of BC, BEAR, CQL, BCQ on these datasets from [55] directly. For the results of TD3+BC and IQL, we run them on these datasets with their official codebases. All methods are run over 4 different random seeds. The hyperparameters we adopt for these tasks are shown in Table 10. We observe that MCQ behaves fairly well on maze2d datasets, and is competitive to prior methods on Adroit tasks. BC and CQL behaves well on Adroit datasets. Note that we find that BC beats MCQ on hammer-human-v0 and pen-cloned-v0. The reason may be attributed to the high complexity and narrow distribution of the Adroit tasks which makes it hard for $Q$ networks to learn well, and the sampled action can easily lie outside of the dataset's support. Nevertheless, MCQ achieves the highest average score on these datasets.

Table 9: Normalized score comparison of different baseline methods on D4RL benchmarks. 0 corresponds to a random policy and 100 corresponds to an expert policy.

| Task Name | BC | BEAR | CQL | BCQ | TD3+BC | IQL | MCQ (ours) |
|---|---|---|---|---|---|---|---|
| maze2d-umaze | -3.2 | 65.7 | 18.9 | 49.1 | 25.7±6.1 | 65.3±13.4 | 81.5 ±23.7 |
| maze2d-umaze-dense | -6.9 | 32.6 | 14.4 | 48.4 | 39.7±3.8 | 57.8±12.5 | 107.8 ±3.2 |
| maze2d-medium | -0.5 | 25.0 | 14.6 | 17.1 | 19.5±4.2 | 23.5±11.1 | 106.8 ±38.4 |
| maze2d-medium-dense | 2.7 | 19.1 | 30.5 | 41.1 | 54.9±6.4 | 28.1±16.8 | 112.7 ±5.5 |
| Average Above | -2.0 | 35.6 | 19.6 | 38.9 | 35.0 | 37.2 | 102.2 |
| pen-human | 34.4 | -1.0 | 37.5 | 68.9 | 0.0±0.0 | 68.7 ±8.6 | 68.5 ±6.5 |
| door-human | 0.5 | -0.3 | 9.9 | 0.0 | 0.0±0.0 | 3.3±1.3 | 2.3±2.2 |
| relocate-human | 0.0 | -0.3 | 0.2 | -0.1 | 0.0±0.0 | 0.0±0.0 | 0.1 ±0.1 |
| hammer-human | 1.5 | 0.3 | 4.4 | 0.5 | 0.0±0.0 | 1.4±0.6 | 0.3±0.1 |
| pen-cloned | 56.9 | 26.5 | 39.2 | 44.0 | 0.0±0.0 | 35.3±7.3 | 49.4 ±4.3 |
| door-cloned | -0.1 | -0.1 | 0.4 | 0.0 | 0.0±0.0 | 0.5±0.6 | 1.3 ±0.4 |
| relocate-cloned | -0.1 | -0.3 | -0.1 | -0.3 | 0.0 ±0.0 | -0.2±0.0 | 0.0 ±0.0 |
| hammer-cloned | 0.8 | 0.3 | 2.1 | 0.4 | 0.0±0.0 | 1.7 ±1.0 | 1.4 ±0.5 |
| Average Total | 7.2 | 13.9 | 14.3 | 22.4 | 11.7 | 23.8 | 44.3 |

Table 10: Detailed hyperparameters used in MCQ, where we conduct experiments on D4RL maze2d "-v1" datasets and Adroit "-v0" datasets.

| Task Name | weighting coefficient $\lambda$ | number of sampled actions $N$ |
|---|---|---|
| maze2d-umaze | 0.7 | 10 |
| maze2d-umaze-dense | 0.95 | 10 |
| maze2d-medium | 0.9 | 10 |
| maze2d-medium-dense | 0.9 | 10 |
| pen-human | 0.3 | 10 |
| door-human | 0.7 | 10 |
| relocate-human | 0.3 | 10 |
| hammer-human | 0.7 | 10 |
| pen-cloned | 0.9 | 10 |
| door-cloned | 0.3 | 10 |
| relocate-cloned | 0.3 | 10 |
| hammer-cloned | 0.5 | 10 |