# OpenReview forum: "Mildly Conservative Q-Learning for Offline Reinforcement Learning"
_NeurIPS.cc/2022/Conference — NeurIPS 2022 Accept_

### Official Review · Reviewer_fVHB · 2022-06-27

**Rating:** 5
**Confidence:** 4
**Soundness:** 3 good
**Presentation:** 3 good
**Contribution:** 3 good

**Summary:**

This paper proposes using mild conservatism in offline RL to benefit generalization and to avoid overly conservative on OOD actions.
Specifically, this paper develops a Mildly Conservative Bellman (MCB) operator for offline RL, where OOD actions are actively trained and their Q values are actively queried.
Theoretical results under the tabular setting and a practical MCB operator are provided.
Empirically, combining the practical MCB operator with SAC performs well on the D4RL MuJoCo locomotion tasks and when transferring from offline learning to online.

**Questions:**

Below are my questions and concerns on this paper.

1. From Proportion 5, even under the rather strong assumption on the discrepancy between $\hat \mu$ and $\mu$, is it still possible that the OOD actions will Q-values higher than the supremum of the in-distribution Q-values? If so, how the proposed Q-learning-based method avoids the detrimental impact of those "overestimated" OOD actions?
2. Why in the proof of Proportion 4 (Below L72 of Appendix), we have $\\{a_i'\\}^N \sim \mathrm{support}(\mu)$? And the $\hat T_1$ here seems inconsistent with Eq. (9).
3. Why in L183-184 "in-distribution actions are still trained to approximate the optimal batch-constraint Q value"? Is it possible that the target value of such in-distribution actions are inflated by the psedo-label? \
Since Eq. (13) is independent of $a^{ood}$ and $\pi$, will the psedo-target value Eq. (13) collapese $Q(s^{in}, a^{ood})$ at different $a^{ood}$ into a same value, even those in-distribution? If so, how could the proposed the method select out good in-distribution actions?

**Limitations:**

The authors briefly addressed the limitations.
No potential negative societal impacts is discussed.

**Strengths And Weaknesses:**

### Strengths
1. The proposed method is well-motivated by theory.
2. The idea of actively train the Q-values of OOD actions is interesting, though in some sense similar to the CQL paper [1].
3. Experiments are extensive and the proposed method generally performs well.

### Weaknesses
1. Since the proposed method requires behavior cloning (BC), it is doubtful if the proposed method can work well on higher-dimensional and/or non-Markovian datasets where BC can be difficult. Theoretical results such as Proportion 5 requires sufficiently accurate BC, which may not be possible on harder datasets. \
In fact, from Table 8, the proposed method performs less favorable on maze2d datasets compared with other stronger baseline such as OptiDICE [2], and is slightly worse than one-step RL [3] on Adroit dataset.
2. There seems to be discrepancies between the impractical theoretical algorithm and the theoretically less-supported practical algorithm, in particular the practical Loss functions (L167-193).
3. From Table 4 & 9, the proposed method requires per-dataset tuning of the weighting coefficient $\lambda$ on a relatively fine grid, which muds the empirical significance since many of the compared baselines actually *unify* hyperparameters across the MuJoCo datasets, *e.g.*, CQL, IQL and TD3+BC.


\
[1] Kumar, Aviral, et al. "Conservative q-learning for offline reinforcement learning." Advances in Neural Information Processing Systems 33 (2020): 1179-1191.
\
[2] Lee, Jongmin, et al. "Optidice: Offline policy optimization via stationary distribution correction estimation." International Conference on Machine Learning. PMLR, 2021.\
[3] Brandfonbrener, David, et al. "Offline rl without off-policy evaluation." Advances in Neural Information Processing Systems 34 (2021): 4933-4946.

---

> ### Author Response · Authors · 2022-08-02
> **Author Response to Reviewer fVHB (part 1/3)**
>
> Thank you for your insightful comments. We give point-to-point response below. We sincerely hope you can re-evaluate our work based on the updated information. If you have any additional questions, we will be happy to have further discussions.
>
> **Q1: Can MCQ work well on higher-dimensional and/or non-Markovian datasets?**
>
> **A1:** Our empirical evaluation on maze2d and Adroit datasets show that MCQ can exhibit good performance on these datasets, where learning a good generative model can be difficult. Compared to some *common* baselines, MCQ achieves the highest average score over all datasets. We thank the reviewer for mentioning OptiDICE [1] and One-step RL [2]. We will cite these papers in our revision. But we think that it is unfair to compare to some strong baselines on some specific datasets. It is quite natural that some methods work well on some specific datasets, e.g., TD3+BC behaves well on MuJoCo dataset but fails on Adroit datasets, OptiDICE shows good performance on maze2d datasets but is not satisfying on MuJoCo datasets, etc. It is also very hard for a single algorithm to outperform all other strong baselines on all types of the datasets. We think that our selected baselines are reasonable as they can typically represent different categories of offline RL algorithms, such as CQL (value penalization method), TD3+BC (policy constraint method), IQL (that learns without querying OOD actions), etc.
>
> [1] Lee, Jongmin, et al. OptiDICE: Offline policy optimization via stationary distribution correction estimation. ICML 2021.
>
> [2] Brandfonbrener, David, et al. Offline rl without off-policy evaluation. NeurIPS 2021.
>
> **Q2: Under the assumption of Proposition 5, is it still possible that the Q-values of OOD actions are higher than the supremum of the in-distribution Q-values?**
>
> **A2:** It is an interesting question. In Proposition 5, we require that $D\_{TV}(\hat{\mu}(\cdot|s)||\mu(\cdot|s))\le \epsilon<\frac{1}{2}$. Such assumption generally requires that the empirical density distribution fits well the true behavior policy. We want to note here that $D\_{TV}(\hat{\mu}(\cdot|s)||\mu(\cdot|s))\in[0,1]$. Then ensuring that $D\_{TV}(\hat{\mu}(\cdot|s)||\mu(\cdot|s))<\frac{1}{2}$ can be satisfied for most situations as CVAE fits the behavior policy in many datasets well in practice. Under such assumption and based on the theoretical results in Proposition 5, the pseudo target value has a chance to exceed $\max\_{a\in\rm{support}(\mu)}Q(s,a)$. However, that does not indicate that bad OOD actions will be executed in practice. The reasons lie in two aspects: (1) the theoretical bound is an *upper* bound, and it does not necessarily mean that the pseudo target value will exceed $\max\_{a\in\rm{support}(\mu)}Q(s,a)$; (2) if the learned behavior policy (CVAE) fits well the true behavior policy, most of the sampled actions from the density model (CVAE) will be in-distribution that are well-trained, i.e., they will not exceed $\max\_{a\in\rm{support}(\mu)}Q(s,a)$. If OOD actions are involved in the actions sampled from the CVAE, its negative impact can be *averaged* and mitigated by these in-distribution actions. Therefore, the pseudo target values for the OOD actions sampled from the trained policy will not be overwhelmed by the overestimated values.
>
> Empirically, we find MCQ exhibits good performance on non-expert datasets and behaves fairly well on expert datasets, which we believe can ease this concern to some extent.
>
> **Q3: Issues on the proof of Proposition 4.**
>
> **A3:** Thanks for pointing that out. That ought to be $\\{a\_i^\prime\\}^N\sim\rm{support}(\hat{\mu})$. That is, the sampled actions are from the fitted behavior policy $\hat{\mu}$, which is then consistent with the definition of $\hat{\mathcal{T}}_1$. We apologize for the typo and will revise it in our revision.

---

> > ### Author Response · Authors · 2022-08-02
> > **Author Response to Reviewer fVHB (part 2/3)**
> >
> > **Q4: Why in L183-184 "in-distribution actions are still trained to approximate the optimal batch-constraint Q value"? Is it possible that the target value of such in-distribution actions are inflated by the psedo-label?**
> >
> > **A4:** For the OOD actions, we actively train them by assigning them pseudo target values. While for in-distribution actions, they are still trained by leveraging the standard bellman error (which approximates the optimal Q-value). Recall that the loss function for MCQ gives below:
> > $$
> > \mathcal{L}\_{\rm{critic}} = \lambda \mathbb{E}_{(s,a,r,s^\prime)\sim\mathcal{D}}[(Q\_{\theta\_i}(s,a)-y)^2] + (1-\lambda)\mathbb{E}\_{s^{\rm{in}}\sim\mathcal{D},a^{\rm{ood}}\sim\pi}[(Q\_{\theta\_i}(s^{\rm{in}},a^{\rm{ood}}) - y^\prime)^2],
> > $$
> > where $y$ is the target value for the in-distribution actions and $y^\prime$ is the pseudo target value for OOD actions.
> >
> > For the case that there exist in-distribution actions in the sampled actions from the trained policy, the corresponding Q-value will approximate $\lambda y + (1-\lambda)y^\prime$ (by taking derivatives w.r.t. Q). We see that this is a convex combination of in-distribution target value and pseudo target value. Then by assigning large weight coefficient $\lambda$, which is suggested in our experiments, we can still make the Q-values of in-distribution actions approximate the optimal batch-constraint Q-value. Meanwhile, as we explained above, the sampled actions from the CVAE are mostly in-distribution, and will not incur severe overestimation. The target value of the in-distribution actions are therefore less likely to be inflated by the pseudo target values. Note that with the auxiliary loss, the OOD actions sampled from the trained policy are projected into the identical pseudo target value, i.e., $y^\prime$. While for in-distribution actions, they approximates the convex combinations of in-distribution target value and OOD pseudo target value. Therefore, the Q-values of the in-distribution actions will not be identical to the Q-values of OOD actions. In this way, MCQ can still select out good in-distribution and safe actions.
> >
> > **Q5: The practical implementation of the method diverges from the theory**
> >
> > **A5:** We would like to argue that many offline RL algorithms also have discrepancies between the theoretical algorithm and the theoretically less-supported practical algorithm, e.g., BCQ [1]. BCQ involves convex combination of double critics when calculating the target values, and also incorporates a perturbation network to increase the diversity of the generated actions. They both lack theoretical support and diverge from the theory of BCQ. It is hard to keep the theoretical algorithm unchanged when combined with deep neural networks. Nevertheless, our key intuition and innovations are consistent, i.e., we actively train OOD actions by assigning them proper pseudo target values. In practice, it is hard to determine whether an action lies in the OOD region (and it is still an open problem). We thus regularize the actions sampled from the learned policy equally. To balance the training of in-distribution samples and OOD actions, we introduce a weighting coefficient $\lambda$, which we believe is a natural and practical solution for applying the practical MCB operator.
> >
> > [1] S. Fujimoto, D. Meger, and D. Precup. Off-Policy Deep Reinforcement Learning without Exploration. ICML 2018.

---

> > > ### Author Response · Authors · 2022-08-02
> > > **Author Response to Reviewer fVHB (part 3/3)**
> > >
> > > **Q6: The proposed method requires per-dataset tuning of the weighting coefficient**
> > >
> > > **A6:** The weighting coefficient $\lambda$ is a vital hyperparameter for MCQ, which balances the training of in-distribution actions and OOD actions. MCQ exhibits superior performance on non-expert datasets, which significantly outperforms IQL, TD3+BC, and CQL. Unfortunately, there is no free lunch. Nevertheless, we experimentally find that $\lambda\in[0.7,1)$ can generally guarantee good performance. When practically applying MCQ, one ought not to use small $\lambda$ as we always want a comparatively large weight on in-distribution actions (i.e., standard bellman error). We demonstrate this by conducting empirical experiments on 3 *random* datasets and 6 *medium* datasets, where we search over $\lambda\in\\{0.0,0.1,0.3,0.5,0.7,0.9\\}$. We observe that the performance of MCQ drops and can hardly learn useful policies with a small $\lambda$, while a large $\lambda$ works fairly well.
> > >
> > > | Task Name | $\lambda=0$ | $\lambda=0.1$ | $\lambda=0.3$ | $\lambda=0.5$ | $\lambda=0.7$ | $\lambda=0.9$ |
> > > | ---- | :---: | :---: | :---: | :---: | :---: | :---: |
> > > | halfcheetah-random | 2.2$\pm$0.6 | 4.6$\pm$1.3 | 5.5$\pm$0.9 | 6.3$\pm$2.6 | 19.5$\pm$0.6 | 27.2$\pm$0.9 |
> > > | hopper-random | 0.7$\pm$0.0 | 1.2$\pm$0.5 | 10.8$\pm$12.1 | 31.0$\pm$0.7 | 31.4$\pm$0.4 | 29.4$\pm$4.3 |
> > > | walker2d-random | -0.1$\pm$0.0 | -0.1$\pm$0.0 | 0.2$\pm$0.2 | 8.7$\pm$7.3 | 14.4$\pm$7.4 | 4.5$\pm$1.3 |
> > > | halfcheetah-medium | -0.3$\pm$0.3 | 38.2$\pm$0.9 | 41.4$\pm$0.7 | 43.9$\pm$0.5 | 49.8$\pm$0.4 | 61.2$\pm$0.3 |
> > > | hopper-medium | 1.7$\pm$0.9 | 21.5$\pm$5.5 | 27.1$\pm$6.9 | 56.7$\pm$18.6 | 78.4$\pm$4.3 | 48.6$\pm$13.2 |
> > > | walker2d-medium | -0.1$\pm$0.1 | 1.5$\pm$1.2 | 60.1$\pm$12.2 | 68.3$\pm$2.8 | 72.8$\pm$5.8 | 91.0$\pm$0.4 |
> > > | halfcheetah-medium-replay | -1.6$\pm$3.7 | 17.6$\pm$3.8 | 38.3$\pm$0.5 | 40.9$\pm$2.0 | 41.3$\pm$1.7 |  55.1$\pm$2.0 |
> > > | hopper-medium-replay | 1.8$\pm$0.8 | 2.3$\pm$2.1 | 3.7$\pm$3.2 | 4.9$\pm$2.6 | 80.7$\pm$20.4 | 101.6$\pm$0.8 |
> > > | walker2d-medium-replay | -0.2$\pm$0.1 | 0.0$\pm$0.2 | 0.3$\pm$0.6 | 1.2$\pm$1.0 | 32.2$\pm$30.9 | 91.3$\pm$5.7 |
> > >
> > > Table 1. Normalized average score of MCQ over different choices of $\lambda$ on MuJoCo "-v2" datasets. The results are averaged over 4 different random seeds.
> > >
> > > **Q7: Connections to CQL**
> > >
> > > **A7:** We want to argue here that MCQ is different from CQL. The main differences lie in: (1) CQL penalizes the Q-values of the actions sampled from the learned policy and maximizes the Q-values of the in-distribution actions; while MCQ **assigns pseudo target values for the OOD actions** such that they can be properly and actively trained. (2) CQL injects too much conservatism into the policy learning, while MCQ reserves "mild" conservatism as the Q-values of the OOD actions are not penalized to be small. (3) MCQ exhibits much better performance than CQL when transferring from offline to online.

---

> > > > ### Comment · Reviewer_fVHB · 2022-08-06
> > > > **Response to the authors**
> > > >
> > > > Dear authors,
> > > >
> > > > Thank you for your detailed response, which address several of my previous questions/concerns.
> > > >
> > > > The following questions/concerns remains.
> > > >
> > > > > Under the assumption of Proposition 5, is it still possible that the Q-values of OOD actions are higher than the supremum of the in-distribution Q-values?
> > > >
> > > > The response to this question does not convince me. Proposition 5 appears on Section 3.2 where the CVAE structure has not been introduced.
> > > > This question/concern is to challenge the authors on the theory/method developmemt, not on the practical implementation via CVAE.
> > > >
> > > > As I discussed in the original/main review, there is a siginificant gap between the theory/method and the actual implementation.
> > > > I have no doubt that the actual implementation **works well under sufficient fine-tuning**. But does that really come from the proposed method and the theoretical insights?
> > > >
> > > > As a side note, it is well-known that CVAE can exhibit a mode-covering behavior, as shown in Section 2.4 of [1].
> > > > Therefore, there is no guarantee that CVAE fits the behavior policy well, not to mention the theoretical assumption that $D_{\mathrm{TV}} < \frac{1}{2}$.
> > > >
> > > > [1] Yang, Shentao, et al. "A Regularized Implicit Policy for Offline Reinforcement Learning." arXiv preprint arXiv:2202.09673 (2022).
> > > >
> > > > > we actively train OOD actions by assigning them proper pseudo target values ... natural and practical solution for applying the practical MCB operator.
> > > >
> > > > I think this paragraph summarize this paper pretty well. In fact, without the theory part, the proposed method can be understood intuitively per you described.
> > > >
> > > > In this regard, why do we need the theory part that cannot be implemented in practice and that does not add to the understanding of the actual implementation?
> > > > For example, the theory part seems to assume knowing the support of the in-distribution actions, which as you discussed "is still an open problem" and is circumvented by "regularize the actions sampled from the learned policy equally" .
> > > >
> > > > > The proposed method requires per-dataset tuning of the weighting coefficient
> > > >
> > > > Based on my calculation, in Table 1 of the response (part 3/3), the average score of $\lambda = 0.7$ variants on the listed nine datasets is $46.7$, $\lambda = 0.9$ variants $56.7$, while the per-dataset tuned variant in Table 1 of the manuscript is $62.3$.
> > > > In this regard, per-dataset tuning of hyperparameters is "vital" for the empirical results of the proposed method, since unifying hyperparameter setting will lead to siginificant drop of the performence.
> > > > This may not align with "offline RL", whose purpose is to learn good policy *without* enviromental interaction.
> > > >
> > > > Further, as I discussed in the original/main review, the significance of the experimeltal results in this paper is harmed by the per-dataset tuning of the weighting coefficient on a fine grid.
> > > > Many of the compared baselines actually unify hyperparameters across the MuJoCo datasets, e.g., CQL, IQL and TD3+BC.
> > > > In this regard, the comparison between the proposed method and the baselines seems unfair.
> > > > Therefore, it is hard to judge the empirical effectiveness of the proposed implementation, which also deviates from the theoretical analysis and is thus less supported.

---

> > > > > ### Author Response · Authors · 2022-08-07
> > > > > **Further clarifications to Reviewer fVHB (part 1/2)**
> > > > >
> > > > > We appreciate the reply from the reviewer and are happy to have a further discussion with the reviewer. We give point-to-point clarifications below, which we hope can mitigate your concerns. If there are still some unaddressed questions, please let us know!
> > > > >
> > > > > **Q1: On Proposition 5 and CVAE implementation**
> > > > >
> > > > > **A1:** We appreciate the recommended paper [1] from the reviewer. There may exist some situations, e.g., the dataset is highly multi-modal, then we can simply replace the CVAE as the conditional GAN (CGAN) to better capture the different modes in the dataset. As depicted in [1], CGAN is beneficial for multi-modal datasets (though it seems GAN with the "state-action joint matching strategy" can best fit the toy eight Gaussian dataset, we think CGAN also exhibits good performance). Whereas for the assumption on the fitted behavior policy, our assumption requires a comparatively well-fitted $\hat{\mu}$. In most cases, CVAE can already fit the dataset well and guarantee a good performance. While for some hard cases, we can adopt CGAN for the density model. We will cite the recommended paper in our revision and add some notes on how to deal with multi-modal datasets (or when CVAE fails). We expect a good empirical behavior policy such that most of the actions sampled from it will be in-distribution, which help average and mitigate the negative possible overestimation impacts from OOD actions.
> > > > >
> > > > > As for the reviewer's concern on whether CVAE can work well in practice, we want to clarify that many algorithms that leverage CVAE, e.g., BCQ [2], PLAS [3], exhibit very good performance on complex non-Markovian datasets like Adroit (please refer to Table 5 in [3]). Our MCQ also shows good performance on maze2d and Adroit datasets. We believe all the evidence can mitigate some concerns on the feasibility of the CVAE, i.e., CVAE can lead to good performance on many datasets (and especially MuJoCo as shown in our experimental results). And if CVAE fails, one can replace it with CGAN or other generative models.
> > > > >
> > > > > We also would like to clarify here that ***our key intuition and innovations are consistent*** for both theoretical forms and practical algorithms, i.e., we construct and assign pseudo target values for OOD actions. Our practical implementation is motivated by our theoretical analysis of the MCB operator. The introduced auxiliary loss significantly boosts the performance of the base SAC algorithm, which indicates that the performance improvement comes from the theoretical insights of MCQ.
> > > > >
> > > > > [1] Yang, Shentao, et al. A Regularized Implicit Policy for Offline Reinforcement Learning. ArXiv.
> > > > >
> > > > > [2]  S. Fujimoto, D. Meger, and D. Precup. Off-Policy Deep Reinforcement Learning without Exploration. ICML.
> > > > >
> > > > > [3] Zhou, W., Bajracharya, S., & Held, D. PLAS: Latent Action Space for Offline Reinforcement Learning. CoRL.
> > > > >
> > > > > **Q2: Why do we need the theory part?**
> > > > >
> > > > > **A2:** Thanks for the comment. As we discussed above, the intuition of our MCQ algorithm comes from the theoretical analysis on the tabular MDP setting. The theoretical analysis provides basic insights and foundations for our proposed auxiliary loss. We always follow the practical application of our MCB operator in the paper. For the initial version of the MCB operator, we cannot directly utilize it since it may be intractable to acquire the maximum over a continuous action space, and the behavior policy is often unknown. Then, we propose the practical MCB operator, where we fit an empirical behavior policy $\hat{\mu}$ and construct the pseudo target values based on it. We present theoretical analysis on the practical MCB operator in Proposition 4 and 5. Furthermore, we extend the practical MCB operator into the deep RL setting, and propose MCQ algorithm. In deep RL, it is challenging to figure out whether the learned policy will execute OOD actions. We therefore regularize all actions sampled from the learned policy. We deem that the whole logic of our paper is clear. We also note here that we actually *do not assume the prior knowledge* about the support of the in-distribution actions for the practical MCB operator (as we construct the pseudo target values based on the empirical behavior policy).

---

> > > > > > ### Author Response · Authors · 2022-08-07
> > > > > > **Further clarifications to Reviewer fVHB (part 2/2)**
> > > > > >
> > > > > > **Q3: On the weighting coefficient**
> > > > > >
> > > > > > **A3:** We do understand this concern from the reviewer. MCQ exhibits superior performance on many non-expert datasets with the cost of tuning the weighting coefficient $\lambda$, as there is no free lunch. We have discussed on the need of tuning the weighting coefficient $\lambda$ as a limitation of our work. We empirically find that $0.7\le \lambda<1$ can guarantee a good performance, which we believe can provide some aid when applying our MCQ algorithm.
> > > > > >
> > > > > > We want to clarify that offline RL defines the setting of learning without interactions with the environment, while it does not necessarily mean that one needs to unify parameters across all datasets. Due to the limited coverage of datasets, distribution shift, and extrapolation errors, it is hard to say that unifying hyperparameters can always guarantee a good performance when encountering a new unknown dataset. It is actually common and valid that we tune parameters for specific datasets in real-world applications. The role of offline RL is leaned towards providing a pre-trained policy, which is fine-tuned with limited interactions with the environment. Under such a setting, hyperparameter tuning is feasible and necessary to guarantee a good pre-trained policy. Moreover, as we show in the paper, our MCQ exhibits superior offline-to-online fine-tuning performance compared to prior methods thanks to the *mild conservatism*.
> > > > > >
> > > > > > There are also many existing offline RL algorithms that tune their hyperparameters for each dataset. For example, MOPO [1], as a typical model-based offline RL algorithm, tunes its hyperparameters per dataset (please see https://github.com/tianheyu927/mopo/tree/master/examples/config/d4rl). We also follow the author's instruction and tune the parameters of UWAC when reproducing it with its official codebase.
> > > > > >
> > > > > > We hope our responses can address the reviewer's concern. Again, we are willing to have further discussions with the reviewer, and we are grateful for the efforts and time that the reviewer spends to help us improve our paper.

---

> > > > > > > ### Comment · Reviewer_fVHB · 2022-08-08
> > > > > > > **Response to further clarifications from the authors**
> > > > > > >
> > > > > > >
> > > > > > > Dear authors,
> > > > > > >
> > > > > > > Thank you for your timely and detailed response.
> > > > > > >
> > > > > > > The discussion on the CVAE part is now much clear.
> > > > > > > I hope that the authors could add into revision the requirement of an empirically-good fitted behavior policy to mitigate the possible negative overestimation impacts from OOD actions.
> > > > > > >
> > > > > > > I agree with the authors that the whole logic of the paper is clear.
> > > > > > > But when going over the described logic, there are several practical compromise especially when stepping into the practical implementation, which make the theory part less significant.
> > > > > > >
> > > > > > > As an aside, I think the definition of practical MCB operator (L124) requires differentiating the case of $\mu(a \mid s) > 0$ and else. So it requires knowing the support of the behavior policy?
> > > > > > >
> > > > > > > I would like to clarify that there is a misunderstanding to my previous response.
> > > > > > > I did not mean that the authors should "unify parameters across all datasets," but per-dataset tuning the hyperparameters via online evaluation, as in this paper, seems unfair compared with the baselines.
> > > > > > >
> > > > > > > The reference of MOPO seems inappropriate, as it belongs to another realm of *model-based* offline RL, where per-dataset tuning of a small amount of hyperparameters seems common.
> > > > > > > This is different from the baselines in the MCQ paper.
> > > > > > >
> > > > > > > I highly appreciate the efforts the authors put into making this paper better, and will remain my neutral rating for this work.

---

> > > > > > > > ### Author Response · Authors · 2022-08-09
> > > > > > > > **Thanks for your reply!**
> > > > > > > >
> > > > > > > > We thank the reviewer for the kind reply! We think many of the suggestions and comments from the reviewer are of great value to make our paper stronger. We are more than happy to include the discussion part of the CVAE into our revision.
> > > > > > > >
> > > > > > > > We apologize that we misunderstand the comments from the reviewer (we think the reviewer comments that we need to know the support of behavior policy for constructing the pseudo target values). Indeed, we need to know the support of the behavior policy in the practical MCB operator (as we need to examine whether the sampled action from the learned policy is OOD). In practice, it is hard to determine whether the sampled action from the learned policy is OOD, we therefore resort to equally assigning them pseudo target values. We would like to clarify that UWAC tunes its parameter accordingly while the hyperparameters of the baseline methods seems unified (please refer to Section 5.5 in UWAC paper http://proceedings.mlr.press/v139/wu21i/wu21i.pdf). We follow the instructions of hyperparameter setup in the UWAC paper when reproducing it with its official codebase. We leave the automatic tuning of $\lambda$ in MCQ as future work (we believe the concerns of the reviewer can be further mitigated then).
> > > > > > > >
> > > > > > > > We really enjoy having such insightful discussions with the reviewer. Again, thanks for your time and efforts in making our paper better!

---

> ### Author Response · Authors · 2022-08-08
> **Follow up with the Reviewer fVHB**
>
> Dear reviewer fVHB,
>
> We thank the efforts and time you spend in reviewing our work. We really appreciate your thoughtful comments on our manuscript. Hopefully our response has addressed your concerns. As the author-reviewer discussion period is ending soon, we wonder whether there are some remaining concerns or questions. We will be glad to have a further discussion. More discussions and suggestions on further improving our paper are always welcomed!
>
> Best regards,
>
> The authors

---

### Official Review · Reviewer_EAmG · 2022-07-09

**Rating:** 6
**Confidence:** 4
**Ethics Flag:** Yes
**Soundness:** 3 good
**Presentation:** 3 good
**Contribution:** 2 fair

**Summary:**

The paper presents a method for offline reinforcement learning based that involves assigning pseudo-Q-values to OOD values called MCQ. The method's main idea is to modify the Q-targets by detecting the out-of-distribution actions via a density model and assign the Q values to these actions similarly to BCQ by taking a maximum of a Q function over N samples from a density model. The main difference to BCQ is that the authors propose to use this backup operator only for OOD actions instead of all actions and use an actor to recover optimal actions from the modified Q-function. The method is evaluated on severals datasets from D4RL where it outperforms the baselines.

**Questions:**

* Did you try implementing the version of the method that regularizes only OOD actions?
* How well does the method perform on other D4RL datasets (for example, antmaze, adroit and kitchen tasks)?
* What implementations of baselines did you use for offline to online experiments?

**Limitations:**

Right now, the main limitation of this work is limited experimental evaluation. In particular, the method is evaluated only on locomotion tasks using an insufficient number of runs. The paper considers a set of tasks different from the standard tasks used in prior work for offline to online experiments. I will raise my score if these concerns are addressed.

**Strengths And Weaknesses:**

# Strengths
* The method is theoretically sound and practical. Even though all method components have been explored in prior work, the idea of not penalizing the values of OOD actions but using a BCQ-style value estimate to impute the values for these actions is novel.
* The practical version of the algorithm (Eqn. 11) is easy to implement.
* MCB demonstrates good empirical results on a subset of D4RL tasks.
* The paper is well written and easy to follow.

# Weaknesses
* The method can be seen as an extension of BCQ. However, the comparison to BCQ is missing.
* The practical implementation of the method diverges from the theory. In particular, the practical implementation of the method omits OOD evaluation and instead regularizes all actions sampled from the training policy, which are not necessary OOD and, throughout training, will certainly become less OOD which can result in over-penalizing the optimal actions due to value underestimation caused by the BCQ-style operator.
* The method is evaluated only on the locomotion tasks from D4RL, which do not require stitching [1] (dynamical programming).
* Also, the method is evaluated using only 4 seeds which might be insufficient.
* The offline to online experiments miss essential details. In particular, it is unclear how the authors obtained results for other methods. Also, the authors pick a different subset of tasks for offline to online finetuning than considered in the original papers (AWAC, IQL).

[1] RvS: What is Essential for Offline RL via Supervised Learning?
S Emmons, B Eysenbach, I Kostrikov, S Levine

---

> ### Author Response · Authors · 2022-08-02
> **Author Response to Reviewer EAmG (part 1/3)**
>
> Thanks for your detailed and valuable comments. We provide clarification to your questions and concerns as below. If you have any further questions or comments, we will be happy to have further discussions.
>
> **Q1: The comparison to BCQ is missing.**
>
> **A1:** We actually compared our MCQ against BCQ in Appendix D, where we also compare MCQ against other recent baselines like Decision Transformer (DT), MOPO, etc. We defer these comparison to the appendix due to page limit. We attach the comparison results below. We observe that MCQ consistently outperforms BCQ on 14 out of 15 datasets.
>
> | Task Name | BCQ | MCQ (ours) |
> | ---- | :---: | :---: |
> | halfcheetah-random | 2.2 $\pm$0.0 | **28.5$\pm$0.6** |
> | hopper-random | 7.8$\pm$0.6 | **31.8$\pm$0.5** |
> | walker2d-random | 4.9$\pm$0.1 | **17.0$\pm$3.0** |
> | halfcheetah-medium | 46.6$\pm$0.4 | **64.3$\pm$0.2** |
> | hopper-medium | 59.4$\pm$8.3 | **78.4$\pm$4.3** |
> | walker2d-medium | 71.8$\pm$7.2 | **91.0$\pm$0.4** |
> | halfcheetah-medium-replay | 42.2$\pm$0.9 | **56.8$\pm$0.6** |
> | hopper-medium-replay | 60.9$\pm$14.7 | **101.6$\pm$0.8** |
> | walker2d-medium-replay | 57.0$\pm$9.6 | **91.3$\pm$5.7** |
> | halfcheetah-medium-expert | **95.4$\pm$2.0** | 87.5$\pm$1.3 |
> | hopper-medium-expert | 106.9$\pm$5.0 | **111.2$\pm$0.1** |
> | walker2d-medium-expert | 107.7$\pm$3.8 | **114.2$\pm$0.7** |
> | Average Above | 55.2 | **72.8** |
> | halfcheetah-expert | 89.9$\pm$9.6 | **96.2$\pm$0.4** |
> | hopper-expert | 109.0$\pm$4.0 | **111.4$\pm$0.4** |
> | walker2d-expert | 106.3$\pm$5.0 | **107.2$\pm$1.1** |
> | Average Total | 64.5 | **79.2** |
>
> Table 1: Normalized average score comparison of MCQ against BCQ on D4RL benchmarks. 0 corresponds to a random policy and 100 corresponds to an expert policy. The experiments are run on MuJoCo "-v2" datasets over 4 random seeds.
>
> We also want to note here that our method, MCQ, is different from BCQ. The main differences lie in: (1) MCQ is built upon SAC while BCQ is built upon TD3; (2) MCQ properly trains OOD actions by assigning them pseudo target values while BCQ does not; (3) BCQ adds perturbation noise to increase the diversity of actions while MCQ does not.
>
> **Q2: The practical implementation of the method diverges from the theory. Did you try implementing the version of the method that regularizes only OOD actions?**
>
> **A2:** This is an interesting and important question. We would like to argue that many offline RL algorithms have this issue, e.g., BCQ [1], MOPO [2], etc. The practical implementation of BCQ involves convex combination of double critics (in target value calculation), and perturbation noise in actions. The error estimator in MOPO is set to be the maximum standard deviation of the learned models in the ensemble, which also lacks theoretical guarantee and diverges from its theory. The involvement of neural networks makes it hard for us to implement MCQ that follows its original theoretical form.
>
> As for MCQ, if the behavior policy $\mu(\cdot|s)$ is previously known, then we can implement MCQ that exactly follows its theory (i.e., Definition 1). Unfortunately, we often do not have prior knowledge about the data-collecting policy $\mu(\cdot|s)$. We then resort to fitting an empirical distribution $\hat{\mu}(\cdot|s)$, and follows Definition 2 (practical MCB operator). However, we cannot directly apply the practical MCB operator in deep RL since it is challenging to evaluate whether an action is OOD (and we cannot say that the action that does not exist in the batch is OOD, especially for continuous action space). We therefore simply assign pseudo target values for all actions sampled from the trained policy such that OOD actions are properly trained.
>
> The actions sampled from the trained policy may have less probability of being OOD with the increment of training steps, while the risk of being OOD still exists. To mitigate such potential threats, we need to regularize actions sampled from the trained policy. In our experiments, we assign large weighting coefficient $\lambda$ to in-distribution samples, which ensures sufficient training on in-distribution transitions. Empirical success of MCQ on non-expert datasets show that MCQ is less likely to over-penalize the optimal actions.
>
> [1] S. Fujimoto, D. Meger, and D. Precup. Off-Policy Deep Reinforcement Learning without Exploration. ICML 2018.
>
> [2] T. Yu, G. Thomas, L. Yu, S. Ermon, J. Y. Zou, S. Levine, C. Finn, and T. Ma. MOPO: Model-based Offline Policy Optimazation. NIPS 2020.
>
> [3] I. Kostrikov, A. Nair, and S. Levine. Offline Reinforcement Learning with Implicit Q-Learning. ICLR 2022.

---

> > ### Author Response · Authors · 2022-08-02
> > **Author Response to Reviewer EAmG (part 2/3)**
> >
> > **Q3: The method is evaluated only on the locomotion tasks from D4RL**
> >
> > **A3:** To show the effectiveness of MCQ, we additionally compare MCQ against baseline methods on maze2d and Adroit datasets in Appendix H. We attach the comparison results below, where we observe that MCQ is competitive or better than prior methods on these tasks.
> >
> > | Task Name | BC | BEAR | CQL | BCQ | TD3+BC | IQL | MCQ (ours) |
> > | ---- | :---: | :---: |:---: |:---: |:---: |:---: |:---: |
> > | maze2d-umaze | -3.2 | 65.7 | 18.9 | 49.1 | 25.7$\pm$6.1 | 65.3$\pm$13.4 | 81.5$\pm$23.7 |
> > | maze2d-umaze-dense | -6.9 | 32.6 | 14.4 | 48.4 | 39.7$\pm$3.8 | 57.8$\pm$12.5 | 107.8$\pm$3.2 |
> > | maze-medium | -0.5 | 25.0 | 14.6 | 17.1 | 19.5$\pm$4.2 | 23.5$\pm$11.1 | 54.8$\pm$14.1 |
> > | maze-medium-dense | 2.7 | 19.1 | 30.5 | 41.1 | 54.9$\pm$6.4 | 28.1$\pm$16.8 | 33.6$\pm$2.9 |
> > | Average Above | -2.0 | 35.6 | 19.6 | 38.9 | 35.0 | 37.2 | **69.4** |
> > | pen-human | 34.4 | -1.0 | 37.5 |  68.9 | 0.0$\pm$0.0 | 68.7$\pm$8.6 | 68.5$\pm$6.5 |
> > | door-human | 0.5 | -0.3 | 9.9 | 0.0 | 0.0$\pm$0.0 | 3.3$\pm$1.3 | 2.3$\pm$2.2 |
> > | relocate-human | 0.0 | -0.3 | 0.2 | -0.1 | 0.0$\pm$0.0 | 0.0$\pm$0.0 | 0.1$\pm$0.1 |
> > | hammer-human | 1.5 | 0.3 | 4.4 | 0.5 | 0.0$\pm$0.0 | 1.4$\pm$0.6 | 0.3$\pm$0.1 |
> > | pen-cloned | 56.9 | 26.5 | 39.2 | 44.0 | 0.0$\pm$0.0 | 35.3$\pm$7.3 | 49.4$\pm$4.3 |
> > | door-cloned | -0.1 | -0.1 | 0.4 | 0.0 | 0.0$\pm$0.0 | 0.5$\pm$0.6 | 1.3$\pm$0.4 |
> > | relocate-cloned | -0.1 | -0.3 | -0.1 | -0.3 | 0.0$\pm$0.0 | -0.2$\pm$0.0 | 0.0$\pm$0.0 |
> > | hammer-cloned | 0.8 | 0.3 | 2.1 | 0.4 | 0.0$\pm$0.0 | 1.7$\pm$1.0 | 1.4$\pm$0.5 |
> > | Average Total | 7.2 | 13.9 | 14.3 | 22.4 | 11.7 | 23.8 | **33.4** |
> >
> > Table 2: Normalized score comparison of different baseline methods on D4RL benchmarks. 0 corresponds to a random policy and 100 corresponds to an expert policy. The results are averaged over 4 different random seeds.
> >
> > **Q4: The method is evaluated using only 4 seeds which might be insufficient**
> >
> > **A4:** We understand the concern. We run MCQ on MuJoCo datasets for another 4 seeds, yielding a total 8 random seeds, which we believe is comparatively sufficient for reliable evaluation. We summarize the results below. We observe that MCQ exhibits similar performance as reported in the main text.
> >
> > | Task Name | MCQ (4 seeds) | MCQ (8 seeds) |
> > | ---- | :---: | :---: |
> > | halfcheetah-random | 28.5$\pm$0.6 | 28.6$\pm$0.5 |
> > | hopper-random | 31.8$\pm$0.5 | 31.5$\pm$0.7 |
> > | walker2d-random | 17.0$\pm$3.0 | 19.1$\pm$5.1 |
> > | halfcheetah-medium | 64.3$\pm$0.2 | 64.2$\pm$0.3 |
> > | hopper-medium | 78.4$\pm$4.3 | 75.6$\pm$7.4 |
> > | walker2d-medium | 91.0$\pm$0.4 | 89.7$\pm$1.5 |
> > | halfcheetah-medium-replay | 56.8$\pm$0.6 | 56.5$\pm$0.8 |
> > | hopper-medium-replay | 101.6$\pm$0.8 | 101.8$\pm$1.1 |
> > | walker2d-medium-replay | 91.3$\pm$5.7 | 91.2$\pm$4.8 |
> > | halfcheetah-medium-expert | 87.5$\pm$1.3 | 86.4$\pm$2.4 |
> > | hopper-medium-expert | 111.2$\pm$0.1 | 108.5$\pm$4.6 |
> > | walker2d-medium-expert | 114.2$\pm$0.7 | 113.8$\pm$1.9 |
> > | Average Above | 72.8 | 72.2 |
> > | halfcheetah-expert | 96.2$\pm$0.4 | 95.9$\pm$0.6 |
> > | hopper-expert | 111.4$\pm$0.4 | 111.3$\pm$0.6 |
> > | walker2d-expert | 107.2$\pm$1.1 | 107.8$\pm$2.3 |
> > | Average Total | 79.2 | 78.8 |
> >
> > Table 3: Normalized average score of MCQ on D4RL benchmarks. 0 corresponds to a random policy and 100 corresponds to an expert policy. The experiments are run on MuJoCo "-v2" datasets.

---

> > > ### Author Response · Authors · 2022-08-02
> > > **Author Response to Reviewer EAmG (part 3/3)**
> > >
> > > **Q5: The offline to online experiments miss essential details.**
> > >
> > > **A5:** For the offline-to-online experiments, as is stated in the main text (line 267-269), we first train baseline methods (TD3+BC, CQL, etc.) and our MCQ for 1M gradient steps offline, and then perform online fine-tuning for another 100K gradient steps for all of them. The online samples are put into the offline buffer directly, where experiences are sampled for online adaptation. The results of baseline methods are acquired by running their official codebases, i.e., CQL (https://github.com/aviralkumar2907/CQL), TD3+BC (https://github.com/sfujim/TD3_BC), IQL (https://github.com/ikostrikov/implicit_q_learning), AWAC (https://github.com/vitchyr/rlkit). All methods are run over 4 different random seeds. We will add the missing details to the appendix.
> > >
> > > We chose a subset of tasks for offline-to-online fine-tuning different from IQL and AWAC to ensure that our empirical experiments on offline-to-online fine-tuning are consistent to the offline experiments (just like IQL does, where it conducts offline learning in antmaze domain and performs offline-to-online fine-tuning on some datasets from antmaze). Meanwhile, we deem that the offline-to-online fine-tuning is not limited to the datasets that are adopted by previous studies. In our experiments, we observe superior performance of MCQ on non-expert datasets such as random and medium-replay in the offline stage. We then want to show that MCQ can exhibit good generalization capability on these non-expert datasets compared with prior methods when performing offline-to-online fine-tuning. We believe it is reasonable that we utilize *random* datasets and *medium-replay* datasets from D4RL MuJoCo locomotion tasks for such evaluation.

---

> ### Author Response · Authors · 2022-08-08
> **Looking forward to your feedback**
>
> Dear reviewer EAmG,
>
> We first would like to thank the reviewer's efforts and time in reviewing our work. We were wondering if our responses have resolved your concerns. We will be happy to have further discussions with the reviewer if there are still some remaining questions! More discussions and suggestions on further improving the paper are also always welcomed! We sincerely look forward to your kind reply!
>
> Best regards,
>
> The authors

---

> > ### Comment · Reviewer_EAmG · 2022-08-08
> > **Re:**
> >
> > Yes, my concerns have been addressed, I will raise my score accordingly.

---

> > > ### Author Response · Authors · 2022-08-08
> > > **Thanks for raising the score!**
> > >
> > > We are happy that the concerns from the reviewer are addressed. We thank the reviewer for raising the score!

---

### Official Review · Reviewer_7rnm · 2022-07-10

**Rating:** 7
**Confidence:** 4
**Soundness:** 3 good
**Presentation:** 3 good
**Contribution:** 3 good

**Summary:**

Offline RL is a topic of significant interest. One common class of approaches is to learn an action-value function but to enforce that the function is `conservative' so that it does not result in a policy which takes actions that were not in the training data (and therefore of unknown value).

This work introduces a ``mildly'' conservative Bellman operator. In particular, for actions in the support of the behavior policy the operator behaves like standard Bellman operator, but for actions outside the behavior policy support it assumes that value is delta less than an action in the support.

They show that this operator will always result in a conservative Q estimate (that is, it will not over-estimate the value of any action). They then introduce a practical approximation (where the behavior policy is estimated by a CVAE) and test on a set of offline RL control tasks. It performs notably better than prior work on poor demonstrates, but does not consistently outperform TD3+BC when expert demonstrations are available.


**Questions:**

Why do you think TD3+BC seems to be better for expert-level demonstrations (for most tasks)?

The checklist for this paper indicates the code is available for reproducing the experiments but I didn't see a link anywhere to the code?


**Limitations:**

Yes

**Strengths And Weaknesses:**

Strengths:
- Well communicated.

- Principled explanation of an algorithm that empirically performs well on a set of benchmark tasks.

- Offline RL is a topic of significant interest to the community and active research.

Weaknesses:

- Ideally, the algorithm would be tested on a different style of tasks as well (e.g. perhaps Atari), rather than only MuJoCo control tasks.

- This approach does not seem to perform as well when expert level demonstrations are available.

Minor:

I found definition 2 (line 123) confusing since it refers to $\mu(a|s)$ which is stated above you are trying to avoid. It is explained further what the actual "practical" solution is when $\mu$ is not known, but I found this a bit confusing on first read.

---

> ### Author Response · Authors · 2022-08-02
> **Author Response to Reviewer 7rnm**
>
> Thanks for your inspiring and thoughtful comments, and thanks for commenting that our paper is "well communicated". We provide clarification to the concerns below. We hope our responses can address your concerns.
>
> **Q1: The algorithm should be tested on a different style of tasks as well**
>
> **A1:** To show the effectiveness of our MCQ algorithm, we provide additional empirical experiments on other datasets in D4RL, maze2d, and Adroit, in the Appendix H. We attach the comparison results below (one can also refer to Appendix H), where we observe that MCQ outperforms baseline methods on many datasets, and is the best in terms of the average normalized score.
>
> | Task Name | BC | BEAR | CQL | BCQ | TD3+BC | IQL | MCQ (ours) |
> | :--- | :---: | :---: |:---: |:---: |:---: |:---: |:---: |
> | maze2d-umaze | -3.2 | 65.7 | 18.9 | 49.1 | 25.7$\pm$6.1 | 65.3$\pm$13.4 | 81.5$\pm$23.7 |
> | maze2d-umaze-dense | -6.9 | 32.6 | 14.4 | 48.4 | 39.7$\pm$3.8 | 57.8$\pm$12.5 | 107.8$\pm$3.2 |
> | maze-medium | -0.5 | 25.0 | 14.6 | 17.1 | 19.5$\pm$4.2 | 23.5$\pm$11.1 | 54.8$\pm$14.1 |
> | maze-medium-dense | 2.7 | 19.1 | 30.5 | 41.1 | 54.9$\pm$6.4 | 28.1$\pm$16.8 | 33.6$\pm$2.9 |
> | Average Above | -2.0 | 35.6 | 19.6 | 38.9 | 35.0 | 37.2 | **69.4** |
> | pen-human | 34.4 | -1.0 | 37.5 |  68.9 | 0.0$\pm$0.0 | 68.7$\pm$8.6 | 68.5$\pm$6.5 |
> | door-human | 0.5 | -0.3 | 9.9 | 0.0 | 0.0$\pm$0.0 | 3.3$\pm$1.3 | 2.3$\pm$2.2 |
> | relocate-human | 0.0 | -0.3 | 0.2 | -0.1 | 0.0$\pm$0.0 | 0.0$\pm$0.0 | 0.1$\pm$0.1 |
> | hammer-human | 1.5 | 0.3 | 4.4 | 0.5 | 0.0$\pm$0.0 | 1.4$\pm$0.6 | 0.3$\pm$0.1 |
> | pen-cloned | 56.9 | 26.5 | 39.2 | 44.0 | 0.0$\pm$0.0 | 35.3$\pm$7.3 | 49.4$\pm$4.3 |
> | door-cloned | -0.1 | -0.1 | 0.4 | 0.0 | 0.0$\pm$0.0 | 0.5$\pm$0.6 | 1.3$\pm$0.4 |
> | relocate-cloned | -0.1 | -0.3 | -0.1 | -0.3 | 0.0$\pm$0.0 | -0.2$\pm$0.0 | 0.0$\pm$0.0 |
> | hammer-cloned | 0.8 | 0.3 | 2.1 | 0.4 | 0.0$\pm$0.0 | 1.7$\pm$1.0 | 1.4$\pm$0.5 |
> | Average Total | 7.2 | 13.9 | 14.3 | 22.4 | 11.7 | 23.8 | **33.4**
>
> Table 1: Normalized score comparison of different baseline methods on D4RL benchmarks. 0 corresponds to a random policy and 100 corresponds to an expert policy.
>
> **Q2: Why do you think TD3+BC seems to be better for expert-level demonstrations (for most tasks)?**
>
> **A2:** We summarize the performance comparison of our MCQ against TD3+BC on *medium-expert* and *expert* datasets in Table 2. We find that MCQ is actually competitive to TD3+BC on most of the datasets that contain expert demonstrations. MCQ achieves the better average score on 3 out of 6 datasets, and is also better in terms of the mean score. TD3+BC behaves naturally well on expert-level datasets with the aid of the behavior cloning (BC) term (BC itself can behave well on expert datasets). While MCQ can achieve competitive performance against TD3+BC by properly training OOD actions.
>
> | Task Name | TD3+BC | MCQ (ours) |
> | ---- | :---: | :---: |
> | halfcheetah-medium-expert | **90.7$\pm$4.3** | 87.5$\pm$1.3 |
> | hopper-medium-expert | 98.0$\pm$9.4 | **111.2$\pm$0.1** |
> | walker2d-medium-expert | 110.1$\pm$0.5 | **114.2$\pm$0.7** |
> | halfcheetah-expert | **96.7$\pm$1.1** | 96.2$\pm$0.4 |
> | hopper-expert | 107.8$\pm$7 | **111.4$\pm$0.4** |
> | walker2d-expert | **110.2$\pm$0.3** | 107.2$\pm$1.1 |
> | Average | 102.25 | **104.62** |
>
> Table 2. Normalized average score comparison between TD3+BC and MCQ on datasets that contain expert demonstrations.
>
> **Q3: Code for the MCQ**
>
> **A3:** We apologize for missing the code for MCQ. To make sure that our results are reproducible, we include a thorough instructions for implementing MCQ in Appendix C.2 along with the detailed hyperparameter setup. We have also uploaded our anonymous code in https://anonymous.4open.science/r/MCQ-BE79/. Our code will be open-sourced and a formal github link will be added in the manuscript upon acceptance.

---

> > ### Comment · Reviewer_7rnm · 2022-08-06
> > **Thanks**
> >
> > Thanks for your response to my comments.
> >
> > My rating for this paper was already and I remain positive so I leave my rating unchanged.

---

> > > ### Author Response · Authors · 2022-08-07
> > > **Thanks for the positive comments!**
> > >
> > > Thanks for keeping the positive score! We also thank the reviewer for the high-quality and positive comments on our manuscript!

---

### Official Review · Reviewer_5dg8 · 2022-07-11

**Rating:** 6
**Confidence:** 4
**Soundness:** 3 good
**Presentation:** 3 good
**Contribution:** 3 good

**Summary:**

This paper points out the previous offline reinforcement learning methods are too conservative about the out-of-distribution (OOD) actions and instead propose a mild conservative algorithm. It introduces an auxiliary loss term to properly train the value function for OOD actions. The proposed method, MCQ, is shown to outperform previous methods empirically and theoretically proved to behave at least as well as the behavior policy, and has no erroneous overestimation.

**Questions:**

Please see the main review.

**Limitations:**

The authors have addresses the limitations in the paper.

**Strengths And Weaknesses:**

Pros:
This paper is clearly written, and the whole structure is organized and easy to follow. The method is well-motivated and the claims in the paper are all supported by either theoretical analysis or experimental results. Although over-pessimism is not a new problem in offline RL, this method is very innovative and elegant in its processing of OOD actions. In comparison with the baselines, MCQ archives a remarkable improvement on random or medium datasets. The authors also make a careful analysis of the sensitivity of hyperparameters.

Cons:
I want to draw the authors' attention to a recent paper that also addresses the over-pessimism of offline RL algorithms (https://arxiv.org/pdf/2207.02200.pdf), which uses an adaptive method. Besides, I am also confused about the results in Figure 2. It seems that decreasing $\lambda$ is always beneficial to the final performance, and there is no trend that introducing the auxiliary loss can help the final performance. I think the authors should use a small scale of $\lambda$ (0-0.3) to explicitly show the benefits of the auxiliary loss.

---

> ### Author Response · Authors · 2022-08-02
> **Author Response to Reviewer 5dg8**
>
> We thank the reviewer for thinking that our method "is very innovative and elegant in its processing of OOD actions". We also thank the reviewer for the thoughtful comments. We hope our responses below can address your concerns.
>
> **Q1: A recent paper [1] also addresses the over-pessimism of offline RL algorithms with an adaptive method**
>
> **A1:** We thank the reviewer for recommending this interesting paper. We actually notice and pay close attention to this paper upon it is available in ArXiv. Unfortunately, this paper is publicly available on July, 2022, and we cannot get access to it before the submission deadline of this venue. We will cite this paper in our revision.
>
> [1] Ghosh, D., Ajay, A., Agrawal, P., \& Levine, S. Offline RL Policies Should be Trained to be Adaptive. ICML 2022.
>
> **Q2: Show the benefits of the auxiliary loss with a small scale of $\lambda$**
>
> **A2:** As a key component of our MCQ algorithm, the auxiliary loss term is of importance to the final performance of the agent. From Figure 2(a) and 2(b) of the main text, we see that decreasing $\lambda$ will negatively affect the performance of the MCQ.
>
> Recall that the loss function for MCQ gives:
> $$
> \mathcal{L}\_{\rm{critic}} = \lambda \mathbb{E}\_{(s,a,r,s^\prime)\sim\mathcal{D}}[(Q\_{\theta\_i}(s,a)-y)^2] + (1-\lambda)\mathbb{E}\_{s^{\rm{in}}\sim\mathcal{D},a^{\rm{ood}}\sim\pi}[(Q\_{\theta\_i}(s^{\rm{in}},a^{\rm{ood}}) - y^\prime)^2],
> $$
> where $y$ is the target value for the in-distribution samples, and $y^\prime$ is the pseudo target values for the OOD actions.
>
> Hence, $\lambda = 0$ will make MCQ assigns no weight over in-distribution samples. The critic will be trained only with the pseudo target values for the OOD actions, which will corrupts the performance of the agent since no reward information is included in the auxiliary loss. We report the results of $\lambda=\\{0, 0.1, 0.3, 0.5, 0.7, 0.9\\}$ on 6 medium-level datasets and 3 random datasets from D4RL MuJoCo "v2" datasets. The results are shown below, where we observe that a small $\lambda$ is a bad choice for all datasets, especially $\lambda=0$. Ideally, we find that $\lambda\in[0.7,1)$ works fairly well for many of the tasks. As discussed in the main text (line 245-247), small $\lambda$ will make the critic loss term overwhelmed by the OOD actions and in-distribution actions cannot be well trained, which is harmful to the performance of the agent.
>
> | Task Name | $\lambda=0$ | $\lambda=0.1$ | $\lambda=0.3$ | $\lambda=0.5$ | $\lambda=0.7$ | $\lambda=0.9$ |
> | ---- | :---: | :---: | :---: | :---: | :---: | :---: |
> | halfcheetah-random | 2.2$\pm$0.6 | 4.6$\pm$1.3 | 5.5$\pm$0.9 | 6.3$\pm$2.6 | 19.5$\pm$0.6 | 27.2$\pm$0.9 |
> | hopper-random | 0.7$\pm$0.0 | 1.2$\pm$0.5 | 10.8$\pm$12.1 | 31.0$\pm$0.7 | 31.4$\pm$0.4 | 29.4$\pm$4.3 |
> | walker2d-random | -0.1$\pm$0.0 | -0.1$\pm$0.0 | 0.2$\pm$0.2 | 8.7$\pm$7.3 | 14.4$\pm$7.4 | 4.5$\pm$1.3 |
> | halfcheetah-medium | -0.3$\pm$0.3 | 38.2$\pm$0.9 | 41.4$\pm$0.7 | 43.9$\pm$0.5 | 49.8$\pm$0.4 | 61.2$\pm$0.3 |
> | hopper-medium | 1.7$\pm$0.9 | 21.5$\pm$5.5 | 27.1$\pm$6.9 | 56.7$\pm$18.6 | 78.4$\pm$4.3 | 48.6$\pm$13.2 |
> | walker2d-medium | -0.1$\pm$0.1 | 1.5$\pm$1.2 | 60.1$\pm$12.2 | 68.3$\pm$2.8 | 72.8$\pm$5.8 | 91.0$\pm$0.4 |
> | halfcheetah-medium-replay | -1.6$\pm$3.7 | 17.6$\pm$3.8 | 38.3$\pm$0.5 | 40.9$\pm$2.0 | 41.3$\pm$1.7 |  55.1$\pm$2.0 |
> | hopper-medium-replay | 1.8$\pm$0.8 | 2.3$\pm$2.1 | 3.7$\pm$3.2 | 4.9$\pm$2.6 | 80.7$\pm$20.4 | 101.6$\pm$0.8 |
> | walker2d-medium-replay | -0.2$\pm$0.1 | 0.0$\pm$0.2 | 0.3$\pm$0.6 | 1.2$\pm$1.0 | 32.2$\pm$30.9 | 91.3$\pm$5.7 |
>
> Table 1. Normalized average score of MCQ over different choices of $\lambda$ on MuJoCo "-v2" datasets. The results are averaged over 4 different random seeds.

---

> > ### Comment · Reviewer_5dg8 · 2022-08-08
> > **Response to the author**
> >
> > Thank the authors for the response and the additional results.
> >
> > I want to apologize for making a mistake in my review. What I want to say is when $\lambda$ approaches 1 (I regard $1-\lambda$ as $\lambda$ in my original review), the performance seems always increasing which confuses me about the role of the auxiliary loss. This table gives me a more clear view of the trend, where I can tell the optimal $\lambda$ is not at 0.9 for hopper-random, walker2d-random, and hopper-medium.
> >
> > I want to know if the authors could also include the results for $\lambda=1$ in this table. From the current results, simply taking $\lambda=0.9$ gives optimal performances in most cases.

---

> > > ### Author Response · Authors · 2022-08-08
> > > **Response to the Reviewer 5dg8**
> > >
> > > We thank the reviewer for the kind reply. We are happy to include the results of $\lambda=1$ in the table, which can be found below. We want to note here that taking $\lambda=1$ will make our MCQ degenerates into the vanilla SAC (since no weight is assigned to the auxiliary loss), which is why we write $\lambda\in[0.7,1)$ works fairly well instead of $[0.7,1]$. We find that the performance of MCQ degrades with $\lambda=1$ for most of the cases, indicating the benefits and effectiveness of the auxiliary loss term (i.e., actively training OOD actions by assigning them proper pseudo target values).
> > >
> > > For MCQ, we ought to assign a comparatively large weight over in-distribution actions (the bellman error) rather than the auxiliary loss to ensure that they are properly trained. Hopefully this response can address your concern. If there are still any remaining questions, please let us know!
> > >
> > > | Task Name | $\lambda=0$ | $\lambda=0.1$ | $\lambda=0.3$ | $\lambda=0.5$ | $\lambda=0.7$ | $\lambda=0.9$ | $\lambda=1.0$ |
> > > | ---- | :---: | :---: | :---: | :---: | :---: | :---: | :---: |
> > > | halfcheetah-random | 2.2$\pm$0.6 | 4.6$\pm$1.3 | 5.5$\pm$0.9 | 6.3$\pm$2.6 | 19.5$\pm$0.6 | 27.2$\pm$0.9 | 29.7$\pm$1.4 |
> > > | hopper-random | 0.7$\pm$0.0 | 1.2$\pm$0.5 | 10.8$\pm$12.1 | 31.0$\pm$0.7 | 31.4$\pm$0.4 | 29.4$\pm$4.3 | 9.9$\pm$1.5 |
> > > | walker2d-random | -0.1$\pm$0.0 | -0.1$\pm$0.0 | 0.2$\pm$0.2 | 8.7$\pm$7.3 | 14.4$\pm$7.4 | 4.5$\pm$1.3 | 0.9$\pm$0.8 |
> > > | halfcheetah-medium | -0.3$\pm$0.3 | 38.2$\pm$0.9 | 41.4$\pm$0.7 | 43.9$\pm$0.5 | 49.8$\pm$0.4 | 61.2$\pm$0.3 | 55.2$\pm$27.8 |
> > > | hopper-medium | 1.7$\pm$0.9 | 21.5$\pm$5.5 | 27.1$\pm$6.9 | 56.7$\pm$18.6 | 78.4$\pm$4.3 | 48.6$\pm$13.2 | 0.8$\pm$0.0 |
> > > | walker2d-medium | -0.1$\pm$0.1 | 1.5$\pm$1.2 | 60.1$\pm$12.2 | 68.3$\pm$2.8 | 72.8$\pm$5.8 | 91.0$\pm$0.4 | -0.3$\pm$0.2 |
> > > | halfcheetah-medium-replay | -1.6$\pm$3.7 | 17.6$\pm$3.8 | 38.3$\pm$0.5 | 40.9$\pm$2.0 | 41.3$\pm$1.7 |  55.1$\pm$2.0 | 0.8$\pm$1.0 |
> > > | hopper-medium-replay | 1.8$\pm$0.8 | 2.3$\pm$2.1 | 3.7$\pm$3.2 | 4.9$\pm$2.6 | 80.7$\pm$20.4 | 101.6$\pm$0.8 | 7.4$\pm$0.5 |
> > > | walker2d-medium-replay | -0.2$\pm$0.1 | 0.0$\pm$0.2 | 0.3$\pm$0.6 | 1.2$\pm$1.0 | 32.2$\pm$30.9 | 91.3$\pm$5.7 | -0.4$\pm$0.3 |
> > >
> > > Table 1. Normalized average score of MCQ over different choices of $\lambda$ on MuJoCo "-v2" datasets. The results are averaged over 4 different random seeds.

---

> > > > ### Comment · Reviewer_5dg8 · 2022-08-08
> > > > **Response to the additional results**
> > > >
> > > > Thanks for the results and they are very clear. I think this work is solid and I will vote for acceptance.

---

> > > > > ### Author Response · Authors · 2022-08-08
> > > > > **Thanks for the positive comments!**
> > > > >
> > > > > We would like to thank the reviewer for the positive comments on our manuscript! Thanks for thinking that our work is solid and is worth being accepted by NeurIPS!

---

### Meta-Review · Area_Chair_qYgs · 2022-08-24

**Recommendation:** Accept
**Confidence:** Certain

**Metareview:**

All reviewers are generally positive or borderline about this paper. Reviewer's note that the method is theoretically sound and practical to implement. Even though all of the components have been explored previously, the authors combine them in a novel approach that convincingly improves over prior works.

Major concerns have been addressed by the author's response, however, I agree with reviewer fVHB that per dataset tuning of $\lambda$ muddies the comparison with previous approaches that do not do similar. I would encourage the authors to additionally report the best performance with a single setting across datasets to make the comparison clearer.



**Award:**

No

---

### Decision · Program_Chairs · 2022-09-14

Accept